# Spatial sampling in human visual cortex is modulated by both spatial and feature-based attention

**Daniel Marten van Es[1]\*, Jan Theeuwes[1], Tomas Knapen[1,2]\***

[1]Behavioural and Movement Sciences, Vrije Universiteit Amsterdam, Amsterdam, The Netherlands; [2]Spinoza Centre for Neuroimaging, Royal Academy of Sciences, Amsterdam, The Netherlands

**Abstract** Spatial attention changes the sampling of visual space. Behavioral studies suggest that feature-based attention modulates this resampling to optimize the attended feature's sampling. We investigate this hypothesis by estimating spatial sampling in visual cortex while independently varying both feature-based and spatial attention. Our results show that spatial and feature-based attention interacted: resampling of visual space depended on both the attended location and feature (color vs. temporal frequency). This interaction occurred similarly throughout visual cortex, regardless of an area's overall feature preference. However, the interaction did depend on spatial sampling properties of voxels that prefer the attended feature. These findings are parsimoniously explained by variations in the precision of an attentional gain field. Our results demonstrate that the deployment of spatial attention is tailored to the spatial sampling properties of units that are sensitive to the attended feature.

DOI: https://doi.org/10.7554/eLife.36928.001

## Introduction

The resolution of the visual system is highest at the fovea and decreases gradually with increasing eccentricity. But the visual system's resolution is not fixed. Attention can be directed to a location in space and/or a visual feature, which temporarily improves behavioral performance (*Posner et al., 1980*; *Rossi and Paradiso, 1995*; *Found and Müller, 1996*; *Carrasco and Yeshurun, 1998*; *Yeshurun and Carrasco, 1999*; *Kumada, 2001*; *Sàenz et al., 2003*; *Wolfe et al., 2003*; *Theeuwes and Van der Burg, 2007*) at the cost of reduced sensitivity for non-attended locations and features (*Kastner and Pinsk, 2004*; *Pestilli and Carrasco, 2005*; *Wegener et al., 2008*).

Attending a location in space increases activity in units representing the attended location, as shown by both electrophysiological (*Luck et al., 1997*; *Reynolds et al., 2000*) and fMRI studies (*Tootell et al., 1998*; *Silver et al., 2005*; *Datta and DeYoe, 2009*). In addition, spatial receptive fields were shown to shift toward an attended location in macaque MT+ (*Womelsdorf et al., 2006*) and V4 (*Connor et al., 1997*). Using fMRI to measure population receptive fields (pRFs; *Dumoulin and Wandell, 2008*; *Dumoulin and Knapen, 2018*), it was found that pRF shifts induced by spatial attention occur throughout human visual cortex (*Klein et al., 2014*; *Kay et al., 2015*; *Sheremata and Silver, 2015*; *Vo et al., 2017*), a process thought to improve visual resolution at the attended location (*Anton-Erxleben and Carrasco, 2013*; *Kay et al., 2015*; *Vo et al., 2017*). Such *spatial resampling* is understood to be the result of an interaction between bottom-up sensory signals and a top-down attentional gain field (*Womelsdorf et al., 2008*; *Klein et al., 2014*; *Miconi and VanRullen, 2016*).

Feature-based attention, for example directed toward color or motion, selectively increases activity in those units that represent the attended feature, as evidenced by electrophysiological

**\*For correspondence:**
daan.van.es@gmail.com (DME);
tknapen@gmail.com (TK)

**Competing interests:** The authors declare that no competing interests exist.

**eLife digest** Much like digital cameras record images using a grid of tiny pixels, our own visual experience results from the activity of many neurons, each with its own receptive field. A neuron's receptive field is the area of visual space – the scene in front of our eyes – to which that neuron responds. But whereas digital pixels have fixed locations, the receptive fields of neurons do not. If we switch our attention to a different area of the scene in front of us, visual neurons move their receptive fields to cover that area instead. We do not need to move our eyes for this to happen, just the focus of our attention.

Moving receptive fields in this way enables the visual system to generate more detailed vision at the new attended location. Unlike a digital camera, the brain is thus much more than a passive recording device. But does the movement of receptive fields also depend on what we are attending to at a given location? Paying attention to tiny details, for example, might require many receptive fields to move by large amounts to produce vision with high enough resolution.

Van Es et al. have now answered this question by using a brain scanner to measure receptive fields in healthy volunteers. The volunteers focused on different visual features, such as color or motion, and to various visual locations. When the volunteers attended to color, their attention was more tightly focused than when they attended to motion. This might be because processing color requires fine-detail vision, whereas we can detect movement with our attention spread over a larger area. As a result, receptive fields moved more when the volunteers attended to color than when they attended to motion.

Movement of visual receptive fields thus depends on what we attend to, as well as where we focus our attention. This adds to our understanding of how the brain filters the information bombarding our senses. This might lead to better diagnosis and treatment of disorders that include attentional problems, such as autism and ADHD. The results could also help develop artificial intelligence systems that, like the visual system, can process information flexibly to achieve different goals.

DOI: https://doi.org/10.7554/eLife.36928.002

(*Treue and Maunsell, 1996*; *Treue et al., 1999*; *McAdams and Maunsell, 2000*; *Maunsell and Treue, 2006*; *Müller et al., 2006*; *Zhang and Luck, 2009*; *Zhou and Desimone, 2011*), fMRI (*Saenz et al., 2002*; *Serences and Boynton, 2007*; *Jehee et al., 2011*), and behavioral reports (*Sàenz et al., 2003*; *White and Carrasco, 2011*). These studies consistently show that feature-based attention modulates processing irrespective of the attended stimulus's spatial location. In addition, feature-based attention also appears to shift featural tuning curves toward the attended value, as reported by both electrophysiological (*Motter, 1994*; *David et al., 2008*) and fMRI studies (*Çukur et al., 2013*).

The similarity in the effects of feature-based and spatial attention on affected neural units suggests a common neural mechanism for both sources of attention (*Hayden and Gallant, 2005*; *Cohen and Maunsell, 2011*). Yet spatial attention necessitates retinotopically precise feedback (*Miconi and VanRullen, 2016*), whereas feature-based attention operates throughout the visual field (*Maunsell and Treue, 2006*). Studies investigating whether one source of attention potentiates the other generally find that interactions are either nonexistent or very weak at the earliest stages of processing (*David et al., 2008*; *Hayden and Gallant, 2009*; *Patzwahl and Treue, 2009*; *Zhang and Luck, 2009*), but emerge at later stages of visual processing (*Hillyard and Münte, 1984*; *Handy et al., 2001*; *Bengson et al., 2012*; *Ibos and Freedman, 2016*), and ultimately influence behavior (*Kingstone, 1992*; *Kravitz and Behrmann, 2011*; *Leonard et al., 2015*; *White et al., 2015*; *Nordfang et al., 2018*). In addition, the effects of feature-based compared to spatial attention arise earlier in time. This occurs both when only feature-based attention is endogenously cued and subsequently guides spatial attention towards the attended feature's location (*Hopf et al., 2004*), and when both types of attention are endogenously cued (*Hayden and Gallant, 2005*; *Andersen et al., 2011*). This supports the idea that feature-based attention can direct spatial attention toward or away from specific locations containing attended or unattended features (*Cohen and Shoup, 1997*; *Cepeda et al., 1998*; *Burnett et al., 2016*).

The studies mentioned above investigated modulatory effects of feature-based attention on spatial attention by measuring changes in response amplitude (e.g. ERP/firing rate). However, no study to date has investigated the effect of feature-based attention on spatial sampling. Yet, exactly this relationship is predicted by behavioral studies. Especially when attention is endogenously cued, feature-based attention has been argued to influence the spatial resampling induced by spatial attention to optimize sampling of visual features for behavior (*Yeshurun and Carrasco, 1998*; *Yeshurun and Carrasco, 2000*; *Yeshurun et al., 2008*; *Barbot and Carrasco, 2017*). Specifically, these authors suggested that when attending features that are processed by neurons with smaller receptive fields, a greater degree of spatial resampling is required to resolve the required featural resolution at the attended location. In the current study, we put this hypothesis to the test by measuring the brain's representation of space under conditions of differential attention.

Specifically, we measured pRFs under conditions of differential spatial attention (i.e. toward fixation or the mapping stimulus) and feature-based attention (i.e. toward the mapping stimulus's temporal frequency or color content). One important reason for studying the effects of attention to color and temporal frequency is that they are known to be processed at different spatial scales. Specifically, color is generally processed at a finer spatial scale compared to temporal frequency. First, this is a result of color compared to temporal frequency information being processed predominantly by the parvocellular rather than the magnocellular pathways, which in turn pool differentially across visual space (*Schiller and Malpeli, 1978*; *Hicks et al., 1983*; *Denison et al., 2014*). Second, temporal frequency and color are processed across different cortical areas (i.e. MT +compared to hV4; *Liu and Wandell (2005)*; *Brouwer and Heeger, 2009*, *Brouwer and Heeger, 2013*; *Winawer et al., 2010*) that have differential spatial precision (*Amano et al., 2009*; *Winawer et al., 2010*). Third, preference for color compared to temporal frequency is generally greater in the fovea compared to the periphery (*Curcio et al., 1990*; *Azzopardi et al., 1999*; *Brewer et al., 2005*), where receptive fields are generally smaller (*Dumoulin and Wandell, 2008*). The hypothesized relation between spatial scale of attended features and spatial resampling as proposed by the behavioral studies mentioned above thus implies that attending color compared to temporal frequency at a particular location should lead to stronger spatial resampling. In addition, as color and temporal frequency are differentially processed across cortical areas, studying these features allows us to investigate whether modulations of spatial resampling by feature-based attention are specific for areas that prefer the attended feature. We specifically chose temporal frequency and not coherent motion, as coherent motion signals have been shown to influence pRF measurements (*Harvey and Dumoulin, 2016*). It was previously shown that attention can be directed to both feature domains (*Wolfe and Horowitz, 2004*; *Cass et al., 2011*).

We characterized how spatial attention influences the sampling of visual space, and subsequently investigated how feature-based attention modulates this spatial resampling. In addition, an explicit gain-field interaction model allowed us to formally capture the pRF position changes resulting from our attentional manipulations (*Klein et al., 2014*). Finally, we also performed a full-field stimulation experiment which allowed us to relate these attentional modulations to each voxel's bottom-up preference for color and temporal frequency.

In brief, our results show that pRF changes are indeed stronger when attending the stimulus's color compared to temporal frequency content. These modulations occurred similarly throughout the visual system, regardless of an area's bottom-up feature preference. We show that these feature-based attentional modulations can be explained by changes in the precision of the attentional gain field. Together, this confirms the idea that the degree of spatial resampling is dependent on the spatial scale at which attended features are processed.

## Results

For the purpose of our study, we adapted a standard traversing bar-stimulus based retinotopic mapping paradigm (*Dumoulin and Wandell, 2008*). Specifically, our bar stimulus was made up out of many Gabor elements, each with one of two color compositions (blue/yellow or green/magenta) and with particular temporal frequency (high or low). Throughout the duration of a bar pass, participants reported changes in fixation mark luminance, in the predominant Gabor color composition, or in the predominant Gabor temporal frequency (see *Figure 1A* and Materials and methods section for additional details). Importantly, visual stimulation was identical across conditions and only the top-down

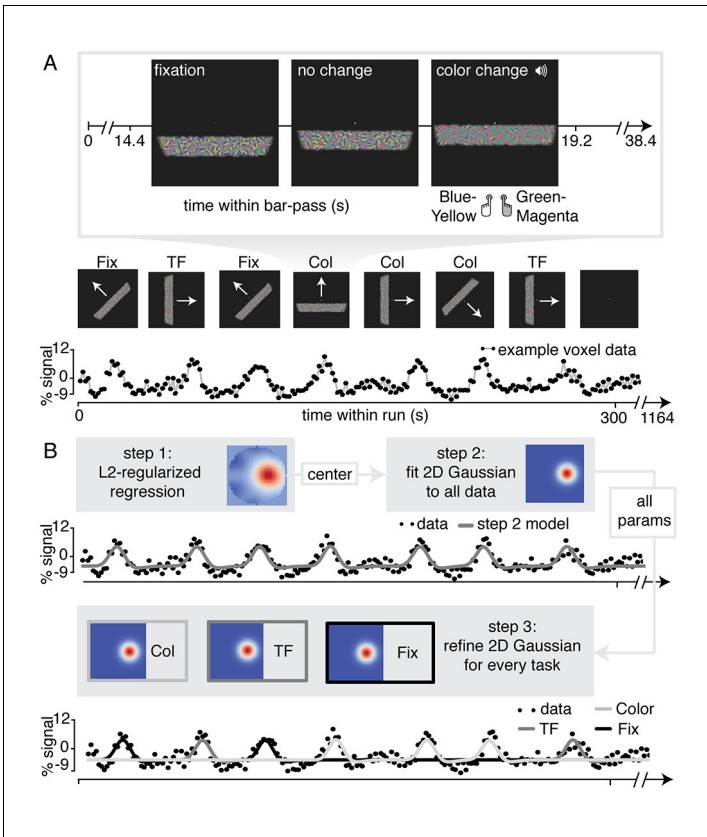

**Figure 1.** Experimental design and pRF fitting procedure. (**A**) Experimental design. Throughout a bar pass participants reported either changes in color (Attend Color) or temporal frequency (Attend TF) of Gabor elements within the moving bar stimulus, or changes in fixation mark luminance (Attend Fixation), while maintaining accurate fixation. Participants were informed auditorily about the upcoming task 2 s before each bar pass. (**B**) Overview of pRF fitting procedure. pRF parameters were estimated from each voxel's BOLD signal time course in a three-step procedure. First, a design matrix was constructed based on 31 × 31 pixels' visual stimulation time course of the entire experiment, which was convolved with a participant-specific HRF (derived from separate data, see Feature preference and HRF Mapper). L2-regularized regression was used to find the position of the spatial response profile's peak of each voxel. Second, to find precise estimates of pRF center location and size, we used a more fine-grained 101 × 101 design matrix and gradient descent to fit a single parameterized 2D Gaussian pRF model to data from all attention conditions combined, initialized at the L2-regression derived peak location. Third, 2D Gaussian pRF models were fitted to data from the different attention conditions separately, initialized with the parameters resulting from step 2.
DOI: https://doi.org/10.7554/eLife.36928.003

factor of attention was manipulated. In addition, behavioral difficulty was kept constant between conditions and across three levels of eccentricity using a Quest staircase procedure. Thus, spatial attention was either directed towards the fixation mark (*Attend Fixation*) or towards the bar stimulus (*Attend Stimulus*). In addition, feature-based attention was either directed towards color (*Attend Color*), or towards temporal frequency (*Attend TF*). This setup allowed us to fit separate population receptive field (pRF) models to data from the different attentional conditions for each voxel (see *Figure 1B* and Materials and methods section for additional details).

Our principal aim is to understand whether feature-based attention influences spatial resampling induced by spatial attention. Therefore, we first characterize in detail the changes in spatial sampling (i.e. pRF properties) that resulted from differential allocation of spatial attention. Second, we capture these pRF modulations by spatial attention in an attentional gain field modeling framework. Third, we investigate how feature-based attention modulated the pattern of pRF changes and how it affected attentional gain field model parameters. Finally, we relate these feature-based attentional

modulations to (1) bottom-up feature preference and (2) to differences in the spatial scale at which color and temporal frequency are processed.

## Region of interest definition

*Figure 2A* shows voxels' *Attend Fixation* location preferences, by depicting color-coded polar angle coordinates on an inflated cortical surface for one example participant's right hemisphere. We examined the relation between pRF eccentricity and size within each of the retinotopic regions, and performed further analyses on those regions that showed clear progressions of polar angle on the surface as well as positive size-eccentricity relations, as shown in *Figure 2B*. In addition, we created

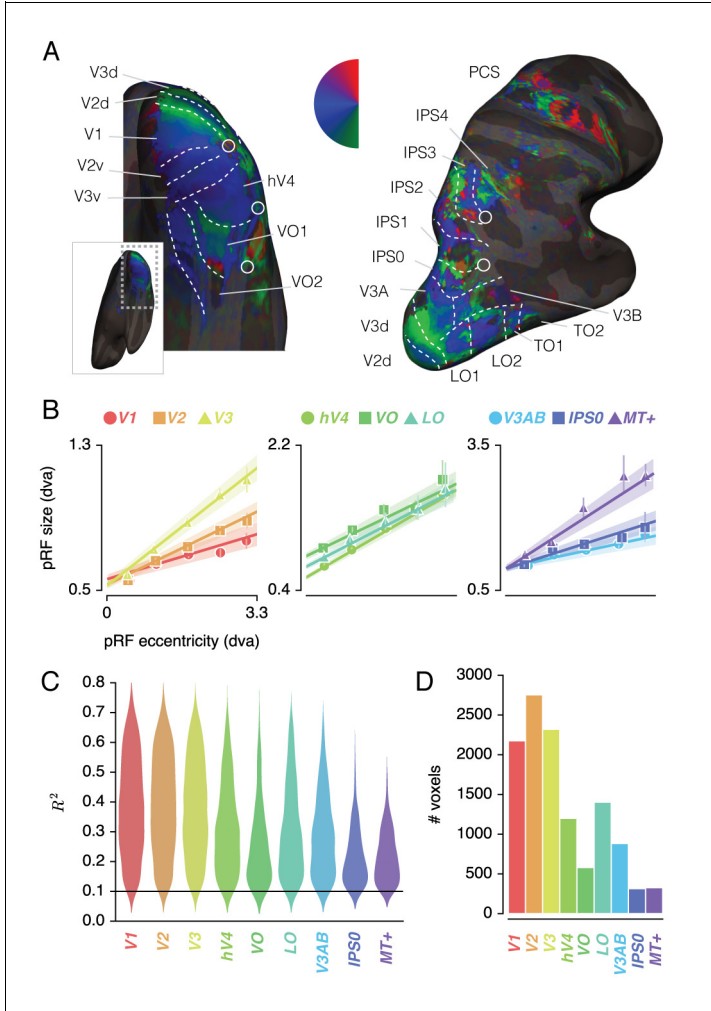

**Figure 2.** ROI definition. (**A**) Attend Fixation pRF polar angle maps for an example participant with retinotopic areas up to the intra-parietal sulcus defined by hand. (**B**) Attend Fixation pRF size as a function of eccentricity for all areas that showed robust relationships. (**C**) Distribution of explained variance across the different regions of interest. The violins are normalized to have equal maximum width. (**D**) Number of voxels per ROI included in all subsequent analyses. All error bars and shaded error regions denote 95% CI of data and linear fits, respectively, across voxels. Also see *Figure 2—figure supplements 1–2*.
DOI: https://doi.org/10.7554/eLife.36928.004

The following figure supplements are available for figure 2:

**Figure supplement 1.** Eccentricity-size relations for all statistical methods.
DOI: https://doi.org/10.7554/eLife.36928.005

**Figure supplement 2.** Explained variance and voxel count for all statistical methods.
DOI: https://doi.org/10.7554/eLife.36928.006

a *combined ROI* that pooled voxels across selected ROIs to evaluate pRF changes across the visual system.

## pRF changes induced by spatial attention

To quantify pRF changes resulting from differential allocation of spatial attention, we created an *Attend Stimulus* condition by averaging pRF parameters between the *Attend Color* and *Attend TF* conditions. To inspect how spatial attention affected pRF positions, we plotted a vector from the *Attend Fixation* to the *Attend Stimulus* pRF position. For visualization purposes, we created visual field quadrant representations by multiplying both the *Attend Fixation* and *Attend Stimulus* pRF x-

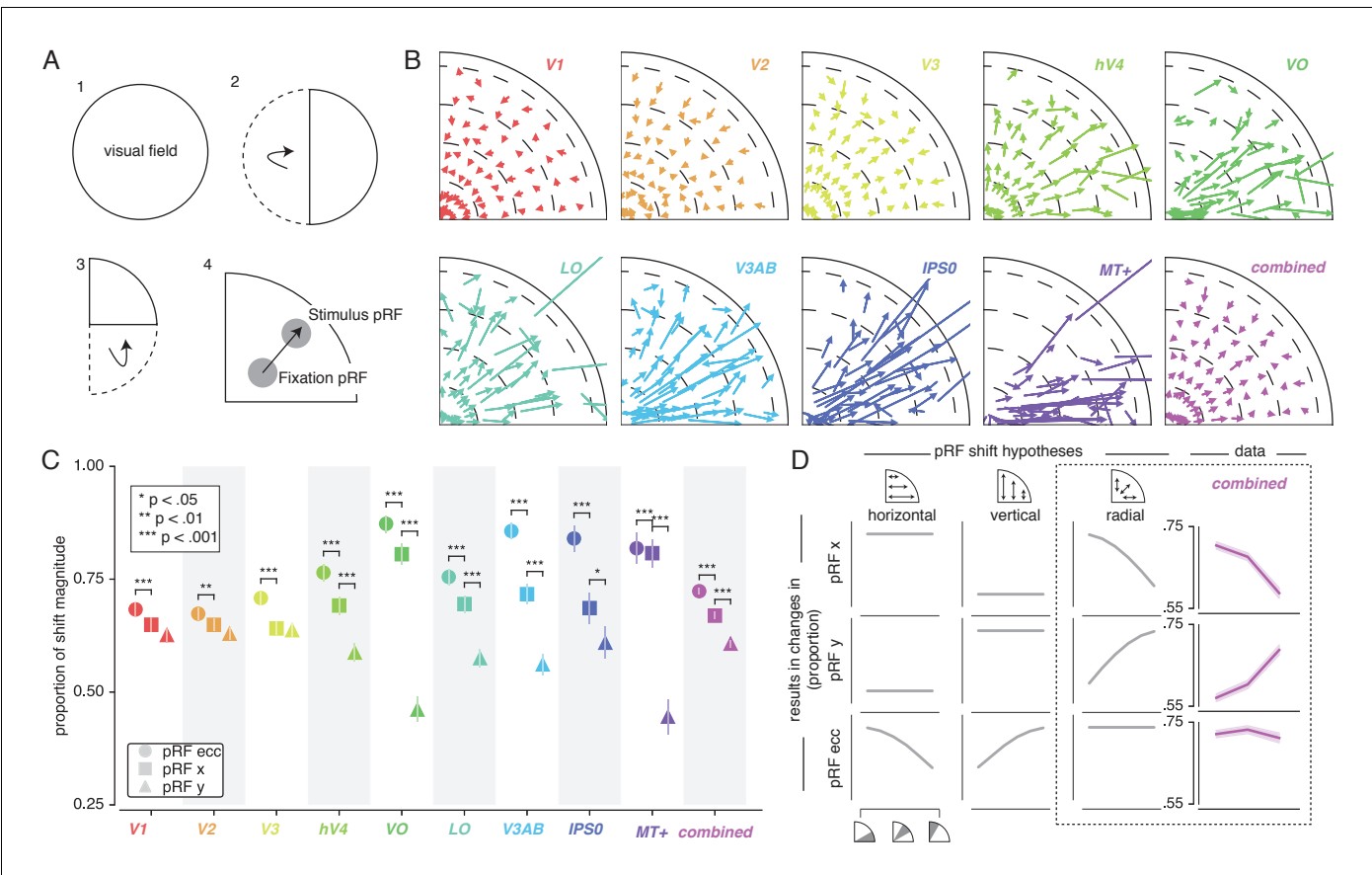

**Figure 3.** Effect of spatial attention on pRF position. (**A**) Plotting strategy. For pRF shift visualizations, all pRF positions are mirrored into one quadrant of the visual field. Then, vectors representing the shift of pRF centers between conditions were drawn from the *Attend Fixation* to the *Attend Stimulus* pRF position. (**B**) Shift vectors as described in (**A**). pRF shift magnitude increased up the visual hierarchy, and shifts appear to occur mainly in the radial direction (i.e. changes in pRF eccentricity). Dotted lines demarcate eccentricity bins used in subsequent analyses. (**C**) Changes in pRF position in the horizontal, vertical and radial directions as a proportion of the length of the shift vectors, as depicted in (**B**). The magnitude of pRF shifts is consistently best described by changes in pRF eccentricity. (**D**) pRF x, y and eccentricity position shifts plotted as a function of polar angle, for different shift direction hypotheses. The data closely match the radial shift direction hypothesis, showing strongest pRF x shifts close to the horizontal meridian, strongest pRF y shifts close to the vertical meridian and strong pRF eccentricity changes across all polar angles. In (**C**), single, double and triple asterisks indicate significant differences with FDR corrected p < 0.05, < 0.01 and < 0.001, respectively. Also see *Figure 3—figure supplements 1–3*.
DOI: https://doi.org/10.7554/eLife.36928.007

The following figure supplements are available for figure 3:

**Figure supplement 1.** pRF shift plots for all statistical methods.
DOI: https://doi.org/10.7554/eLife.36928.008

**Figure supplement 2.** pRF shift directions for all statistical methods.
DOI: https://doi.org/10.7554/eLife.36928.009

**Figure supplement 3.** pRF x, y and eccentricity position shifts as a function of polar angle for all statistical methods.
DOI: https://doi.org/10.7554/eLife.36928.010

and y-coordinates with the sign of pRF x- and y-coordinates in the *Attend Fixation* condition (see *Figure 3A*). This means, for example, that pRFs in the upper-right quadrant were unaffected (i.e. x- and y-coordinate multiplied by 1), while pRFs in the lower-right quadrant were mirrored along the y-axis (i.e. x-coordinate multiplied by 1, y-coordinate multiplied by −1). Note that mirroring based on the *Attend Fixation* condition preserves any pRF shifts across a meridian, allowing pRFs to shift outside the target visual field quadrant. Visual inspection of these pRF position shifts shows both increasing shift magnitude up the visual hierarchy and shifts occurring mainly along the radial dimension (i.e. toward or away from the fovea; *Figure 3B*).

## pRF shift direction

This latter observation seems at apparent odds with a recent study reporting that pRFs shift mainly in the horizontal direction (*Sheremata and Silver, 2015*). To quantify the observed direction of pRF shifts we computed the ratio of shifts in the radial, horizontal and vertical directions (see *Figure 3C*). In line with the data of *Sheremata and Silver (2015)*, we find that changes of pRF horizontal location consistently better describe the overall shifts than do changes of pRF vertical location in all ROIs except V1/2/3 (p's < 0.05, see *Supplementary file 1* -Table 2). We also find that pRF shifts are described even better by shifts in the radial dimension (i.e. changes in eccentricity) compared to shifts in the horizontal direction in all ROIs (p's < 0.01, see *Supplementary file 1* -Table 2). *Figure 3D* is intended to ease interpretation of these results. It depicts how different hypotheses regarding the underlying directionality of pRF shifts (i.e. horizontal, vertical or radial - i.e. foveopetal/foveofugal), translate into changes in measured pRF x, y and eccentricity as a function of quarter visual field polar angle (i.e. from vertical to horizontal meridian). For example, if pRFs shift primarily in the radial direction (right hypothesis column, *Figure 3D*), this would result in the strongest pRF x-direction changes close to the horizontal meridian and the strongest pRF y-direction changes close to the vertical meridian. pRF eccentricity changes however, would show no dependence on polar angle. *Figure 3D*, right column, shows that the data (*combined ROI*) correspond most to the radial shift hypothesis. To quantify this visual intuition, we compared the slopes of the change in pRF $x$ and $y$ over polar angle by binning polar angle into three bins and comparing the first and last bins (i.e. horizontal and vertical meridian, respectively). This showed that, compared to the slope of pRF $y$ change over polar angle, the slope of pRF $x$ change was more negative (p < 0.001, Cohen's d = 0.677, N = 11946). This pattern of results can only be explained by pRFs shifting in the radial direction. Visual field coverage is known to be non-uniform such that the horizontal meridian is overrepresented at both subcortical (*Schneider et al., 2004*) and cortical (*Swisher et al., 2007*; *Silva et al., 2018*) levels and was also clearly present in our data (Rayleigh tests for non-uniformity in ROIs separately, p's < 0.001, see *Supplementary file 1* -Table 3). This means that shifts that occur exclusively in the radial dimension appear as a dominance of horizontal compared to vertical shifts when averaging over the visual field.

## pRF changes across eccentricity

To further inspect the attention-induced radial shifts described above, we plotted the difference between *Attend Stimulus* and *Attend Fixation* pRF eccentricity for each of four *Attend Fixation* pRF eccentricity bins (*Figure 4A*). This returned varying patterns of pRF eccentricity changes across the different ROIs. The *combined ROI* shows that overall, parafoveal pRFs shifted away from the fovea, while peripheral pRFs shifted toward the fovea. These outward shifting parafoveal pRFs are found in all other ROIs except V1 and V2, whereas the inward shifting peripheral pRFs are also present in V1, V2 and V3 (see *Supplementary file 1* -Tables 4 and 5).

In addition to pRF position changes, we also inspected changes in pRF size induced by differences in spatial attention as a function of *Attend Fixation* pRF eccentricity (*Figure 4B*). Overall, parafoveal pRFs increased in size, while peripheral pRFs decreased in size. These expanding parafoveal pRFs were present in all ROIs except V2/3, whereas shrinking peripheral pRFs were found in all ROIs except V1, MT+ and IPS0 (see *Supplementary file 1* -Tables 6 and 7). Overall, this pattern of results is strikingly similar to the changes in pRF eccentricity described above. In fact, the changes in pRF size and eccentricity were strongly correlated in all ROIs (*Figure 4C*, Pearson R over 20 5-percentile bins between 0.74 and 0.99, p's < 0.001, see *Supplementary file 1* -Table 8). Together, these results

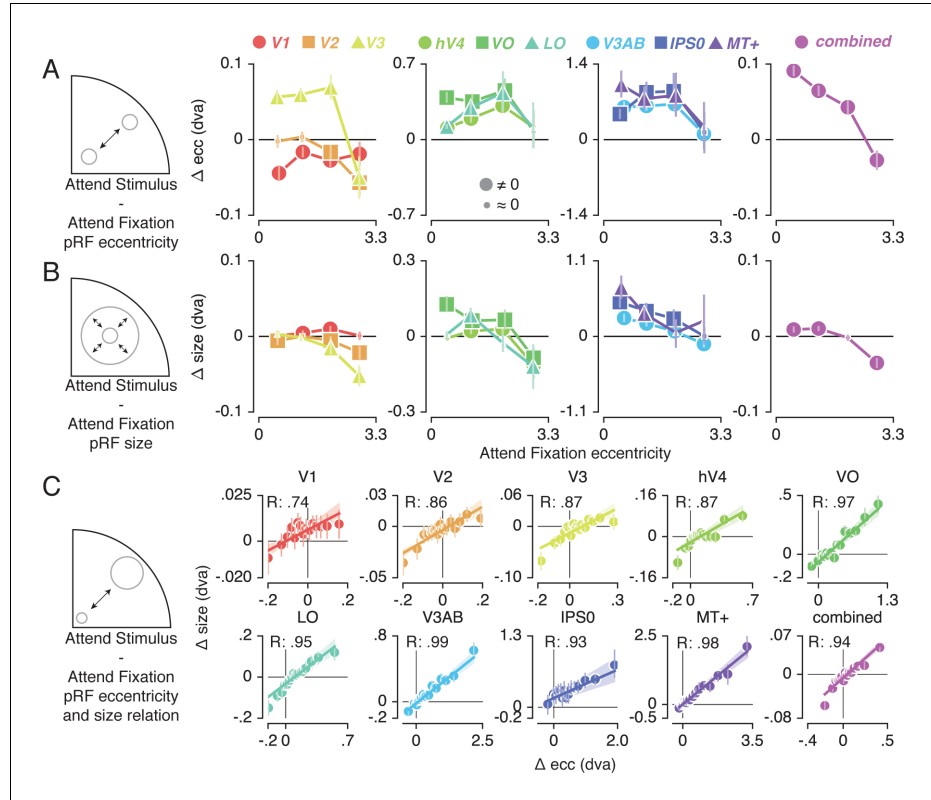

**Figure 4.** Effect of spatial attention on pRF eccentricity and size. Difference between *Attend Stimulus* and *Attend Fixation* pRF eccentricity (**A**) and size (**B**) as a function of *Attend Fixation* eccentricity. Overall, parafoveal pRFs tend to shift away from the fovea and increase in size, while peripheral pRFs tend to shift toward the fovea and decrease in size. (**C**) Changes in pRF eccentricity and size were strongly correlated in all ROIs. In (**A**) and (**B**), markers are increased in size when bootstrapped distributions differ from 0 with FDR corrected p < 0.05. In (**C**) the markers' error bar denotes 95% CI of data over voxels and shaded error regions denote 95% CI of linear fit parameters over bins. Also see *Figure 4—figure supplements 1–2*.

DOI: https://doi.org/10.7554/eLife.36928.011

The following figure supplements are available for figure 4:

**Figure supplement 1.** Effect of spatial attention on pRF eccentricity and size for all statistical methods.
DOI: https://doi.org/10.7554/eLife.36928.012

**Figure supplement 2.** Correlation between pRF eccentricity and size changes for all statistical methods.
DOI: https://doi.org/10.7554/eLife.36928.013

show that attention to the stimulus caused parafoveal pRFs to shift away from the fovea and increase in size, whereas peripheral pRFs shifted toward the fovea and decreased in size.

## Formal account for observed pattern of pRF shifts

To provide a mechanistic explanation for the complex pattern of pRF shifts described above, we modeled our results using a multiplicative Gaussian gain field model (*Womelsdorf et al., 2008*; *Klein et al., 2014*). We adapted this framework to work in conditions where attention shifted over space as a function of time (see Materials and methods). In brief, this modeling procedure used the *Attend Fixation* pRF, one attentional gain field at fixation and another convolved with the stimulus to predict the *Attend Stimulus* pRF position. We determined optimal attentional gain field sizes by minimizing the difference between observed and predicted *Attend Stimulus* pRF positions in the quadrant visual field format of *Figure 3B*. *Figure 5A* illustrates that model predictions closely followed the data, thereby accurately reproducing radially shifting pRFs. Examining the predicted change in pRF eccentricity as a function of eccentricity (i.e. the dominant pRF shift direction; *Figure 5B*) showed that the model was able to capture widely varying eccentricity change profiles

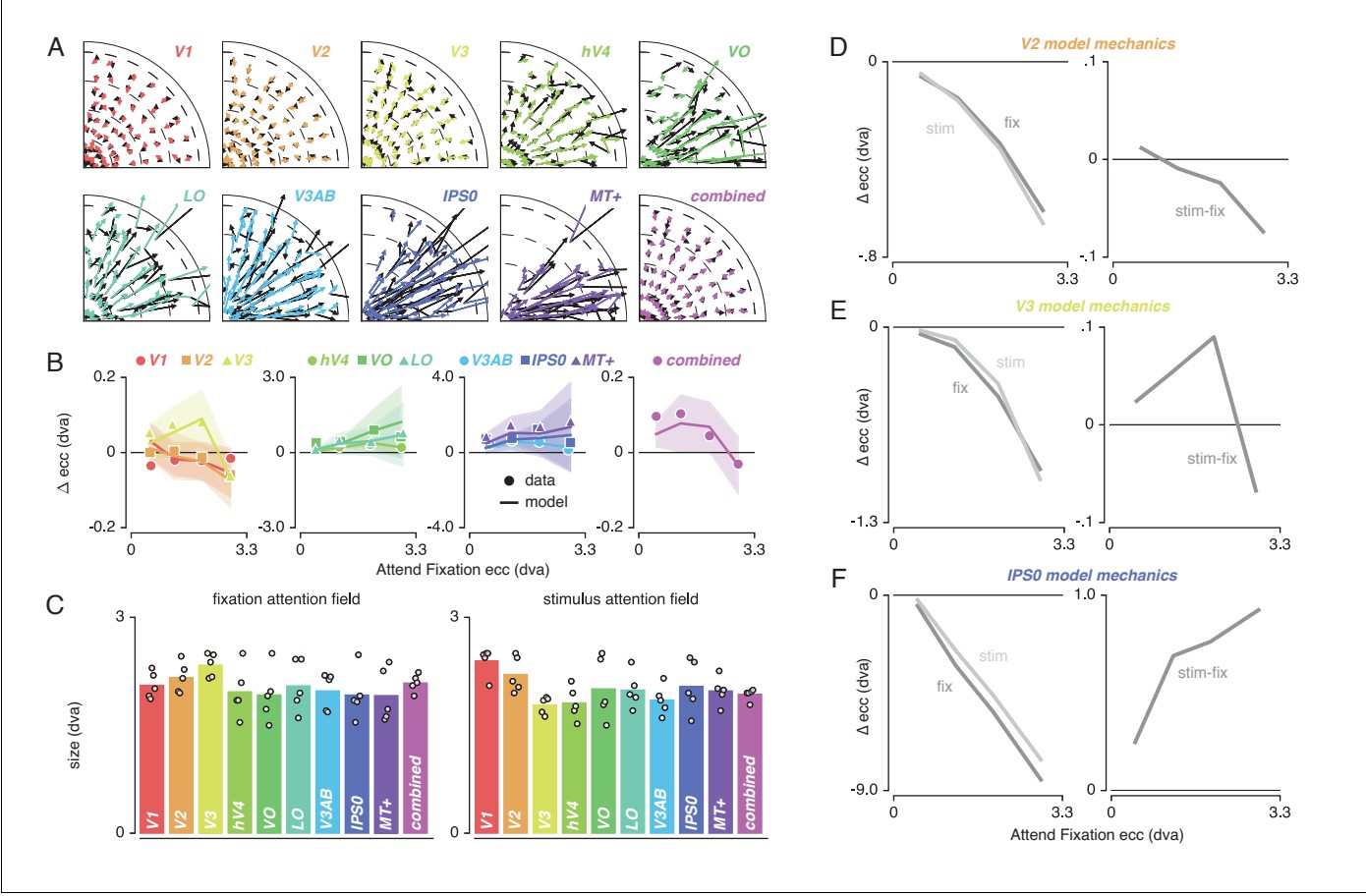

**Figure 5.** Capturing spatial attention in attentional gain field model. (A) Observed (black) and predicted (color) pRF shifts. (B) Observed and predicted changes in pRF eccentricity (the main pRF shift direction) as a function of eccentricity. Markers depict data and lines the corresponding attentional gain field model fit. (C) Fitted fixation and stimulus attentional gain field sizes. Dots depict individual subjects, and bars the average across subjects. (D-F). Left panels depict changes in eccentricity induced by attending fixation (dark gray lines) and by attending the stimulus (light gray lines). Although both sources of attention cause a pull toward the fovea in all ROIs, relative shift magnitude differs across eccentricity. The right panels show how the difference between both spatial attention conditions results in the patterns as observed in (B). In (B), error bars denote 95% CIs over subjects. Plotting conventions as in *Figure 3*. Also see *Figure 5—figure supplements 1–2*.

DOI: https://doi.org/10.7554/eLife.36928.014

The following figure supplements are available for figure 5:

**Figure supplement 1.** Attentional gain field modeling results for each subject - shift vectors.

DOI: https://doi.org/10.7554/eLife.36928.015

**Figure supplement 2.** Attentional gain field modeling results for each subject - changes in pRF eccentricity.

DOI: https://doi.org/10.7554/eLife.36928.016

across ROIs using very similar attentional gain field sizes (*Figure 5C*). This shows that a common attentional influence can result in very different pRF shift patterns, which then depend on differential spatial sampling properties across ROIs (i.e. distribution of pRF sizes and positions). In sum, these results show that the attentional gain field model provides a parsimonious and powerful account for the variety of pRF shift patterns across ROIs.

We further investigated how the model was able to reproduce the eccentricity-dependent eccentricity changes we reported above. For this, we inspected pRF shifts induced by attending either fixation or the stimulus relative to the stimulus drive (i.e. the pRF outside the influence of attention derived from the model). For illustrative purposes, we display results for V2, V3 and IPS0 as these areas showed marked differences in their eccentricity change profile (*Figure 5D–F*). The left panels of each figure reveal the effects of attending fixation and the stimulus separately. This shows that both sources of spatial attention pull the measured pRFs toward the fovea, albeit with differing

relative magnitudes across eccentricity. The right panel of each figure shows that the resulting difference between attending fixation and the stimulus constitutes the eccentricity-dependent patterns observed in the data (*Figure 5B*).

Together, these analyses show that existing multiplicative gain field models of attention can be extended to predict pRF shifts in situations where spatial attention shifts over time. Additionally, it confirms, extends and quantifies earlier reports showing that the precision of the attentional gain field is similar across the visual hierarchy (*Klein et al., 2014*).

## Feature-based attentional modulation

Having established (1) the pattern of changes in spatial sampling (i.e. changes in pRF size and eccentricity) resulting from differential allocation of spatial attention, and (2) a mechanistic explanation of these changes, we next examined how this pattern was modulated by differences in feature-based attention. *Figure 6A* shows how pRF eccentricity and size are differentially affected by attending color or temporal frequency within the stimulus for the *combined ROI*. This illustrates that while both tasks caused similar pRF changes, these effects were generally more pronounced when attending color.

To quantify the modulation of feature-based attention per voxel, we first set up a single robust index of the degree to which spatial attention resampled visual space, combining changes in pRF eccentricity and size (as these were highly correlated, see *Figure 4C*). This Attentional Modulation Index (AMI, see Materials and methods) is depicted in *Figure 6B* for the *combined ROI* when attending color and TF. We then quantified the difference in this AMI between attending color and temporal frequency as a feature-based Attentional Modulation Index (feature AMI, see Materials and methods). Positive values of feature AMI indicate that attending color induced greater pRF changes, while negative values indicate that attending TF led to stronger pRF changes. *Figure 6C* shows that this feature AMI was positive across eccentricity in the *combined ROI*. Inspecting the average feature AMI across voxels within each ROI (*Figure 6D*) reveals that attending changes in color compared to TF in the bar stimulus produced stronger spatial resampling in all ROIs (p's < 0.01, see *Supplementary file 1* -Table 9). Specifically, the feature AMI was around 0.05 on average across ROIs. As the feature AMI is a contrast measure where difference is divided by the sum, this corresponds to roughly 10% stronger pRF changes when attending color compared to temporal frequency. In some ROIs (V3AB/IPS0, see *Figure 6—figure supplement 1*), the FAMI reached values of 0.20, which corresponds roughly to 50% stronger pRF changes when attending color compared to temporal frequency. Computing the AMI with either pRF eccentricity or size changes separately (i.e. not as a combined measure) yields similar results (see *Figure 6—figure supplement 2*).

## Feature AMI and feature preference

The feature-based modulations we describe above are possibly related to differences in bottom-up preference for the attended features. Feature-based attention is known to increase activity of neurons selective for the attended feature, regardless of spatial location (*Treue and Maunsell, 1996*; *Treue et al., 1999*; *McAdams and Maunsell, 2000*; *Maunsell and Treue, 2006*; *Müller et al., 2006*; *Zhang and Luck, 2009*; *Zhou and Desimone, 2011*). This suggests that if activity in a given voxel is modulated more strongly by changes in a certain feature, this could lead to a greater apparent pRF shift when attending that feature. To test this hypothesis, we estimated the difference in response amplitude to the presence of color and temporal frequency within a full-field stimulus (in a separate experiment, see Materials and methods). We then summarized each voxel's relative preference for color and temporal frequency by means of a feature preference index. Higher values of this feature preference index indicate greater preference for color compared to TF. *Figure 6E* displays the feature AMI as a function of feature preference, for each ROI. Note that feature preference was negative on average in most ROIs, suggesting that our TF manipulation (7 vs 0 Hz grayscale Gabors) caused stronger response modulations compared to our color manipulation (colored vs grayscale Gabors). Although this induced an offset across the brain, variations in this measure across ROIs replicate known specializations of the visual system with high precision (*Liu and Wandell, 2005*; *Brouwer and Heeger, 2009*; *Brouwer and Heeger, 2013*): while areas MT+ and V1 show the strongest preference for TF compared to color, areas V4 and VO show the strongest preference for color

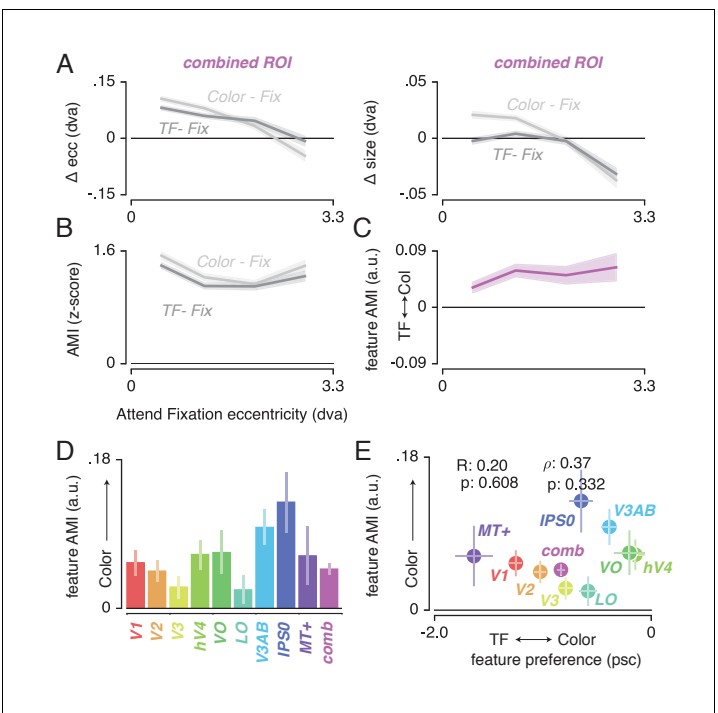

**Figure 6.** Feature-based attentional modulation of pRF changes. (**A**) Differences in pRF eccentricity and size relative to the *Attend Fixation* condition, for both the *Attend Color* and *Attend TF* conditions. The changes in both eccentricity and size are more pronounced when attending changes in color versus TF changes in the bar. (**B**) The Attentional Modulation Index (AMI) combines eccentricity and size changes to form one robust index of spatial attention and is greater when attending color compared to TF. (**C**) The feature AMI quantifies this difference. Positive values of this feature AMI across eccentricity confirm stronger pRF modulations when attending color compared to TF. (**D**) Average feature AMI for each ROI, extending greater observed pRF modulations when attending color compared to TF to all individual ROIs. (**E**) Average feature AMI as a function of average feature preference across ROIs. Feature preference increases with higher color compared to TF preference. Although hV4 and VO are relatively sensitive to color and MT+ is relatively sensitive to TF, feature AMI is comparable in these areas. Error bars denote 95% CI over voxels. Also see *Figure 6—figure supplements 1–4*.
DOI: https://doi.org/10.7554/eLife.36928.017

The following figure supplements are available for figure 6:

**Figure supplement 1.** Feature-based attentional modulation of pRF parameters across eccentricity for each ROI.
DOI: https://doi.org/10.7554/eLife.36928.018

**Figure supplement 2.** Feature AMI compared to feature preference across ROIs with different ways of computing the FAMI.
DOI: https://doi.org/10.7554/eLife.36928.019

**Figure supplement 3.** Feature-based attentional modulation of pRF size and eccentricity changes in the *combined* ROI for all statistical methods.
DOI: https://doi.org/10.7554/eLife.36928.020

**Figure supplement 4.** Feature AMI compared to feature preference for each ROI, for all statistical methods.
DOI: https://doi.org/10.7554/eLife.36928.021

compared to TF. Importantly, regardless of these large variations in feature preferences between MT+/V1 and VO/hV4, average feature AMI was nearly equal in these ROIs. In addition, there was no correlation between feature preference and feature AMI across all ROIs (R = 0.20, p = 0.608, N = 9, rho = 0.37, p = 0.332, N = 9). These results show that the observed feature-based attentional modulations occur globally across the brain, and do not depend on bottom-up feature preference.

## Feature preference and spatial sampling
How do we explain that attending color in the stimulus induced greater changes in spatial sampling? Behavioral studies have suggested that the influence of spatial attention should be adjusted by

feature-based attention to improve sampling of attended visual features (*Yeshurun et al., 2008*; *Barbot and Carrasco, 2017*). Specifically, these authors suggested that attending features processed by relatively smaller receptive fields requires a greater degree of spatial resampling. This implies that if color-preferring voxels are relatively small, this could explain the greater degree of resampling observed when attending color. This is indeed predicted by the fact that color compared to temporal frequency information is predominantly processed by the parvocellular compared to the magnocellular pathway, where spatial sampling is generally more fine-grained (*Schiller and Malpeli, 1978*; *Hicks et al., 1983*; *Denison et al., 2014*). In addition, pRF size varies on average between visual regions, such that pRF size is generally larger in area MT+ (preferring temporal frequency) compared to hV4 (preferring color). Finally, both pRF size (*Dumoulin and Wandell, 2008*) and color compared to TF preference (*Curcio et al., 1990*; *Azzopardi et al., 1999*; *Brewer et al., 2005*) are known to be strongly eccentricity-dependent such that foveal voxels have relatively small pRFs and are relatively color-sensitive. We also clearly observe both effects in our data (see *Figure 2B* and *Figure 7*, correlation between feature-preference and eccentricity is negative except in LO and VO, see *Supplementary file 1* - Table 10). In sum, the greater amount of spatial resampling when attending color can be parsimoniously explained by color being sampled by relatively smaller pRFs.

## Feature-based attention influences attentional gain field precision

Smaller pRFs also require a more precise attentional gain field to shift a given distance (a property of the multiplication of Gaussians). Combining this with our observation that pRFs experience greater shifts when attending color, we predict that attentional gain fields should be more precise in this condition. To test this, we repeated the attentional gain field modeling procedure described above, replacing the *Attend Stimulus* data with the *Attend Color* and *Attend TF* data in two separate fit procedures. Both our data (*Figure 5*) and previous findings (*Klein et al., 2014*) showed that a single attentional gain field affects the different visual regions similarly. In addition, our results presented above showed that pRF modulation by feature-based attention was not related to feature preference across ROIs. We therefore analyzed feature-based attentional modulations of the attentional gain field both on data from all ROIs fitted together (the 'combined ROI'), and as the median across individually fitted ROIs. This analysis returned smaller fitted stimulus attentional gain field sizes in the *Attend Color* compared to the *Attend TF* fit procedure (*Figure 8*) both in the *combined ROI* (0.094 dva smaller over subjects when attending color, $t(4) = 9.021$, $p = 0.001$, Cohen's $d = 4.511$) and across ROIs (median over ROIs on average 0.061 dva smaller over subjects when attending color, $t(4) = 4.243$, $p = 0.013$, Cohen's $d = 2.121$). As the *Attend Fixation* data were used as input in both modeling procedures, we verified that the estimated fixation attentional gain field was not different between procedures (p's of .693 and .224 and Cohen's d of $-0.213$ and $-0.719$ for across ROIs and *combined ROI*, respectively). These analyses show that the stronger influence of spatial attention when attending color is realized by a more precise attentional gain field located at

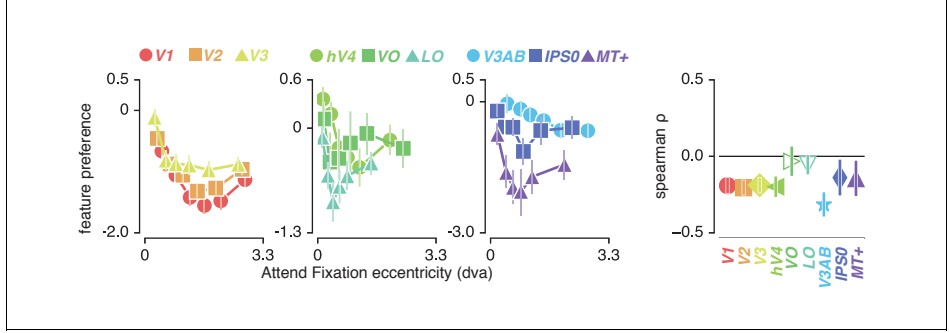

**Figure 7.** Feature preference and eccentricity. Preference to color compared to TF is greatest near the fovea. Error bars denote 95% CI over voxels. Also see *Figure 7—figure supplement 1*.
DOI: https://doi.org/10.7554/eLife.36928.022

The following figure supplement is available for figure 7:

**Figure supplement 1.** Color compared to TF preference versus eccentricity correlations for all statistical methods.
DOI: https://doi.org/10.7554/eLife.36928.023

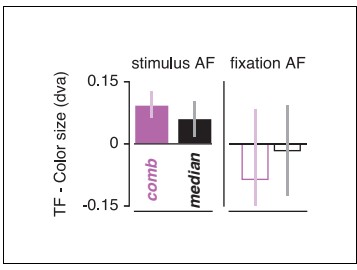

**Figure 8.** Feature-based attentional modulation of attentional gain field sizes. Results are shown for the combined ROI and for the median across ROIs. Positive values here indicate smaller attentional gain fields when attending color compared to temporal frequency in the bar. This means that positive values demarcate stronger spatial resampling when attending color (in correspondence with *Figure 6D and E*). Error bars represent 95% CI over subjects.

DOI: https://doi.org/10.7554/eLife.36928.024

the stimulus. In sum, our results suggest that (1) the attentional system adjusts its influence in accordance with the spatial sampling characteristics of units that prefer the attended feature and (2) that it does this equally across visual regions regardless of their bottom-up feature preference.

## Across-subject consistency

To evaluate the stability of our results, we repeated all analyses for individual subjects. The figures and details for these results can be found in the supplements of the specific figures, and in additional statistical tables. This showed that although spatial attention resulted in somewhat different patterns of pRF shifts between subjects, these individual differences were well captured by the attentional gain field model. This suggests that individual differences in pRF changes are likely the result of the known individual differences in pRF parameter distributions (e.g. eccentricity-size relations [*Figure 2—figure supplement 1*, and see *Dumoulin and Wandell, 2008*]). In addition, these analyses showed that spatial resampling was consistently modulated by feature-based attention across subjects (i.e. feature AMI was on average 0.059 greater than 0 ($F_{(1,4)}$ = 18.868, p = 0.012, $\eta^2 p$ = 0.394), and was not different between ROIs ($F_{(8,32)}$ = 0.631, p = 0.746, $\eta^2 p$ = 0.066)).

## Task and fixation performance

Finally, we verified that the pRF results were not affected by differences in fixation accuracy or behavioral performance (see *Figure 9*). To provide evidence in favor of these null hypotheses, we performed JZL Bayes factor analyses (using JASP; *Love et al., 2015*) as frequentist statistics are not capable of providing evidence for the null (*Altman and Bland, 1995*). We rotated recorded eye

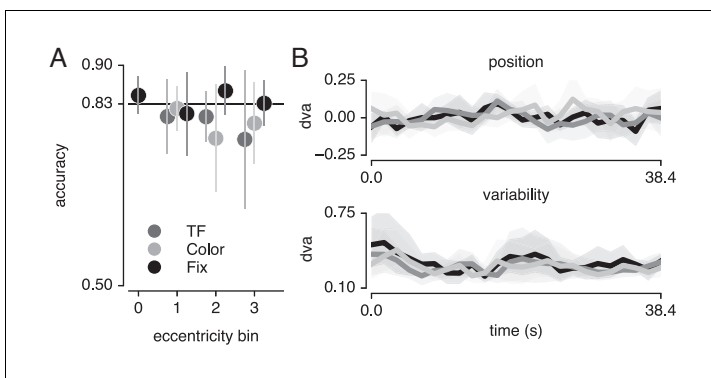

**Figure 9.** Task and fixation performance. (**A**) Behavioral accuracy per attention condition and per bar stimulus eccentricity bin. Horizontal line denotes Quest target of 83%; chance level was 50%. (**B**) Median (top panel) and standard deviation (bottom panel) gaze position in the direction of bar movement per bar position. Error bars denote 95% CI across five participants. Also see *Figure 9—figure supplement 1*.

DOI: https://doi.org/10.7554/eLife.36928.025

The following figure supplement is available for figure 9:

**Figure supplement 1.** Ratio of Gabor elements of either feature value used by the Quest procedure to equate difficulty.

DOI: https://doi.org/10.7554/eLife.36928.026

position to the direction of bar movement and computed the median and standard deviation of position along this dimension across bar passes per bar position (*Figure 9B*). We next set up a model including the factor of attention condition (3 levels), bar position (24 levels) and their interaction. We found that when predicting gaze position, the evidence was in favor of the null hypothesis with a Bayes Factor (BF) of 18620. When predicting gaze variability, however, we found evidence against the null hypothesis with a BF of 5.980. Evidence for including each of the factors (condition, bar position and their interaction) into the model returned BFs of 0.713, 547.193 and 0.017, respectively. The BF of 0.713 for the factor of condition means that we cannot determine whether gaze variability was different between conditions. However, even if this were the case, this could only lead to an offset in pRF size and not to changes in pRF position (*Levin et al., 2010*; *Klein et al., 2014*; *Hummer et al., 2016*). In addition, any anisotropy in gaze position variability could potentially lead to offsets in pRF center positions. Nevertheless, these biases would be identical for all pRFs throughout the visual field. As we find that peripheral pRFs shift inwards and decrease in size and central pRFs shift outwards and increase in size, global offsets in pRF size and position cannot parsimoniously explain our results. More importantly, the analyses also showed that although bar position influenced gaze variability (BF of 547.193), it did not do so differently between attention conditions (BF of 0.017).

Although we used a Quest procedure to equate difficulty across attention conditions and across different levels of eccentricity, it is possible that this procedure stabilized at a faulty difficulty level. To verify whether the Quest procedure successfully equated performance, we used a similar Bayesian approach, testing whether a model including attention condition (three levels) and stimulus eccentricity (three levels) influenced behavioral performance (*Figure 9A*). This returned evidence for the null hypothesis with a BF of 6.25. Together, these results show that differences in pRF parameters between conditions cannot be explained by either fixation accuracy or behavioral difficulty.

## Discussion

We investigated how spatial and feature-based attention jointly modulate the sampling of visual space. We found that directing covert spatial attention toward a moving bar stimulus altered the eccentricity and size of pRFs in concert. These changes in spatial sampling were parsimoniously explained by an attentional gain field model. Attending color changes within this stimulus induced stronger pRF changes compared to attending temporal frequency changes. These feature-based attentional modulations occurred globally throughout the brain, irrespective of a visual region's average feature preference. We suggest that the greater degree of spatial resampling when attending color is related to smaller pRF sizes in relatively color preferring voxels. In addition, we showed that the greater degree of spatial resampling when attending color is caused by a more precise attentional gain field on the stimulus.

Previous behavioral reports suggested that the spatial scale at which an attended feature is processed influenced the degree of spatial resampling (*Yeshurun and Carrasco, 1998*; *Yeshurun and Carrasco, 2000*; *Yeshurun et al., 2008*; *Barbot and Carrasco, 2017*). Specifically, features processed at a finer spatial scale require a greater degree of spatial resampling. This means that the greater degree of spatial resampling when attending color compared to temporal frequency could be explained by color being sampled at a finer spatial scale. This is a canonical difference between parvocellular small, color-sensitive receptive fields and magnocellular large, temporal frequency-sensitive receptive fields (*Schiller and Malpeli, 1978*; *Hicks et al., 1983*; *Denison et al., 2014*). In addition, both our data and previous findings show that color is preferentially processed by visual areas that on average have smaller receptive field sizes (i.e. hV4 compared to MT+ in *Liu and Wandell (2005)*; *Amano et al., 2009*; *Brouwer and Heeger, 2009*, *Brouwer and Heeger, 2013*; *Winawer et al., 2010*). Finally, both the current and previous studies show that pRF size (*Dumoulin and Wandell, 2008*) and color compared to temporal frequency preference (*Curcio et al., 1990*; *Azzopardi et al., 1999*; *Brewer et al., 2005*) vary across eccentricity such that foveal voxels have smaller pRFs and are more color-sensitive. In sum, we suggest that the greater degree of spatial resampling when attending color compared to temporal frequency can be explained by the difference in spatial scale at which these features are processed. We therefore predict that our results should generalize to any other comparison of attended visual features as long as these features differ in their spatial scale. This includes attending different feature values such as

high compared to low spatial frequency, or attending different feature dimensions such as faces (broader spatial scale) versus letters (finer spatial scale).

Electrophysiological studies on the interaction between feature-based and spatial attention generally measure overall response amplitudes rather than changes in spatial sampling. This implies that interactions between feature-based and spatial attention are weak to non-existent in relatively early stages of processing (*David et al., 2008*; *Hayden and Gallant, 2009*; *Patzwahl and Treue, 2009*; *Zhang and Luck, 2009*), but develop at later stages of visual processing (*Hillyard and Münte, 1984*; *Handy et al., 2001*; *Andersen et al., 2011*; *Bengson et al., 2012*; *Ibos and Freedman, 2016*), but see *Egner et al., 2008*). We add to this (1) that feature-based attention modulates the effects of spatial attention on spatial resampling, and (2) that these interactions occur globally throughout the brain, manifesting themselves in even the earliest cortical visual regions. Interactions between spatial and feature-based attention in the early stages of processing could be concealed when focusing on changes in response amplitude rather than changes in spatial sampling. However, it is important to note that measuring spatial sampling at the level of voxels does not allow us to determine whether observed changes in spatial sampling are the result of changes in spatial sampling of individual neurons, or rather the result of differential weighting of subpopulations of neurons within a voxel. Nevertheless, it has been shown that spatial attention does influence spatial sampling of individual neurons (*Connor et al., 1997*; *Womelsdorf et al., 2006*). Yet, future studies are required to extend our conclusions regarding the interactions between feature-based and spatial attention to the single neuron level.

The greater degree of spatial resampling when attending color compared to temporal frequency occurred throughout the brain, irrespective of visual regions' preference for the attended features. In other words, while MT+ and hV4 differed greatly in their relative feature preference, both areas showed comparable pRF changes resulting from differences in feature-based attention. This stands in apparent contrast to previous studies reporting that feature-based attention selectively enhances responses in cortical areas specialized in processing the attended feature (*Corbetta et al., 1990*; *Chawla et al., 1999*; *O'Craven et al., 1999*; *Schoenfeld et al., 2007*; *Baldauf and Desimone, 2014*). However, attending a stimulus consisting of multiple features was shown to spread attentional response modulations of one of the object's feature dimensions to other constituent feature dimensions (*Katzner et al., 2009*; *Çukur et al., 2013*; *Kay and Yeatman, 2017*), albeit somewhat later in time (±60 ms; *Schoenfeld et al., 2014*). This could mean that the global pattern of pRF shifts we observed is caused by such an object-based attentional transfer mechanism. In addition, changing the sampling of visual space globally throughout the brain enhances stability in the representation of space. Different modifications of visual space per visual region would require an additional mechanism linking different spatial representations. Instead, the global nature of spatial resampling we observe supports a temporally dynamic but spatially consistent adaptation of visual space.

An important remaining question pertains to the source of the interactions between feature-based and spatial attention. Signals of spatial selection are thought to originate from a network of frontal and parietal areas, identified using fMRI (*Shulman et al., 2002*; *Silver et al., 2005*; *Jerde et al., 2012*; *Sprague and Serences, 2013*; *Szczepanski et al., 2013*; *Kay and Yeatman, 2017*; *Mackey et al., 2017*) and electrophysiology (*Moore and Armstrong, 2003*; *Gregoriou et al., 2009*). As we focused on careful measurement of spatial sampling in feature-sensitive visual cortex with a relatively small stimulus region, we did not include the frontoparietal regions containing large receptive fields in our analyses. A recent study suggested a central role for the ventral prearcuate gyrus for conjoined spatial and feature-based attentional modulations (*Bichot et al., 2015*). Correspondingly, signals of feature selection in humans have been localized to a likely human homologue of this area, the inferior frontal junction (IFJ; *Zanto et al., 2010*; *Baldauf and Desimone, 2014*). This region is therefore a possible candidate for controlling the interactions between feature-based and spatial attention.

The average changes in pRF size and eccentricity for each visual region in our data are largely consistent with previous studies in which attention was devoted to a peripheral stimulus versus fixation (*Kay et al., 2015*; *Sheremata and Silver, 2015*). Moreover, our analyses go beyond these average pRF changes by investigating the spatial structure of the complex pattern of pRF changes that resulted from such differential spatial attention. The resulting characterization details how the sampling of visual space by single voxel pRFs is affected by spatial attention, which is of specific relevance for future studies that determine spatial selectivity for voxel selections. First, we show that

attending the stimulus compared to fixation caused pRFs to shift radially. Although a previous study reported a dominant horizontal shift direction (*Sheremata and Silver, 2015*), we suggest that the overrepresentation of the horizontal meridian (*Schneider et al., 2004*; *Swisher et al., 2007*) made radially shifting pRFs appear as predominantly horizontal changes. Second, we report closely coupled pRF eccentricity and size changes that were dependent on pRF eccentricity. Specifically, we found that parafoveal pRFs shifted toward the periphery and increased in size, whereas peripheral pRFs shifted toward the fovea and decreased in size. This finding supports the resolution hypothesis of attention (*Anton-Erxleben and Carrasco, 2013*), which posits that spatial attention acts to reduce resolution differences between the fovea and periphery. We note that the functional implication of pRF size changes was recently questioned, as stimulus encoding fidelity was shown to be unaffected by pRF size changes (*Vo et al., 2017*). However, the exact functional significance of changes in pRF size bears no consequence for the conclusions currently presented. As we observed a strong correlation between pRF eccentricity and size changes, we combined both measures into a single robust index. Our relevant quantifications are therefore agnostic to the potentially separable functional implications of changes in pRF size and eccentricity.

The pattern of pRF shifts we observe is described well by an attentional gain field model (*Reynolds and Heeger, 2009*; *Klein et al., 2014*). First, this highlights that a simple and well-understood mechanism underpins the apparent complexity of the observed pattern of pRF changes. Second, it extends the utility of such attentional gain field models to situations in which attention is dynamically deployed over space and time during the mapping of the pRF (*Kay et al., 2015*). In agreement with earlier reports (*Klein et al., 2014*; *Puckett and DeYoe, 2015*), we found that the best-fitting model implemented comparable attentional gain field sizes across visual regions. This strongly points to spatial attention being implemented as a global influence across visual cortex. We conclude that differences in pRF shift patterns between different visual regions depended primarily on differences in visual sampling (i.e. differences in pRF center and size distributions) rather than on differing attentional influences. Despite the broad correspondence between model fits and data, the model did not capture the observed decreases in pRF eccentricity of the most foveal pRFs in V1. Two recent studies showed that in early visual areas, spatial attention shifted pRFs away from the attended location, but toward the attended location in higher visual areas (*de Haas et al., 2014*; *Vo et al., 2017*). Other studies showed that in precisely these visual regions, both the pRF and the attentional gain field are composed of a suppressive surround in addition to their positive peak (*Zuiderbaan et al., 2012*; *Puckett and DeYoe, 2015*). We leave the question of whether these suppressive surrounds could explain such repulsive shifts in lower visual cortex for future research. As a more general aside, gain fields have been shown to influence visual processing at the motor stage (*Van Opstal et al., 1995*; *Snyder et al., 1998*; *Trotter and Celebrini, 1999*). Thus, our results further establish the close link between attentional and motor processes (*Rizzolatti et al., 1987*; *Corbetta et al., 1998*).

In sum, we show that visuospatial sampling is not only affected by attended locations but also depends on the spatial sampling properties of units that prefer attended visual features. The global nature of these modulations highlights the flexibility of the brain's encoding of sensory information to meet task demands (*Rosenholtz, 2016*).

## Materials and methods

### Participants

Five participants (two female, two authors, aged between 25 and 37) participated in the study. All gave informed consent, and procedures were approved by the ethical review board of the University of Amsterdam (2016-BC-7145), where scanning took place.

### Apparatus

#### MRI acquisition

All MRI data were acquired on a Philips Achieva 3T scanner (Philips Medical Systems), equipped with a 32-channel head coil. T1-weighted images were acquired for each participant with isotropic resolution of 1 mm, repetition time (TR) of 8.2 ms, TE of 3.73 ms, flip angle of 8°. Functional T2*-weighted data consisted of 30 2D slices of echo planar images (EPI) with isotropic resolution of 2.5 mm, with a

0.25 mm slice gap, TR of 1600 ms, TE of 27.62 ms, and a flip angle of 70°. Each participant completed between 6 and 8 Attention-pRF Mapping runs (20 min each) and 2–3 Feature preference and HRF Mapping runs (10 min each), spread over 2 (N = 1) or 3 (N = 4) sessions within a 2-week period (see Experimental design).

### Gaze recording
During all functional runs, gaze position was recorded using an Eyelink 1000 (SR Research, Osgoode, Ontario, Canada), sampled at 1000 Hz. A 9-point calibration-validation procedure was run at the start of each session.

### Stimulus presentation
Visual stimuli were created in PsychoPy (*Peirce, 2008*) and run on a 15-inch 2013 MacBook Pro Retina. Participants viewed a 32-inch MR-ready LCD 'BOLDscreen' monitor (resolution: 1920 × 1080, refresh rate: 100 Hz; Cambridge Research Systems), at 156 cm distance from the participants' eyes at the end of the bore, via a helmet-mounted front-silvered mirror. Auditory stimuli were presented through headphones using the MRConfon system.

## Experimental design
### Attention-PRF mapping stimulus
A bar stimulus of 0.9 degrees of visual angle (dva) width traversed a circular aperture of 7.2 dva in one of eight directions (cardinal and diagonal orientations in both directions, see *Figure 1A*), completing a full pass in 38.4 s by stepping 0.34 dva every 1.6 s, and pausing 4.8 s between each bar pass. One run contained 24 bar passes in total (three for every direction), plus four blank periods of 38.4 s when no bar stimulus was shown. Throughout the experiment, a gray disk of 9.6 arcmin (60 cd/m$^2$), with a 4.2 arcmin black rim (0 cd/m$^2$) was present on the screen as a fixation mark.

   Although our stimulus spanned roughly 5% of the visual field (assuming a visual field of 144 dva), this corresponds to roughly 25% of V1 surface area because of cortical magnification (*Adams and Horton, 2003*). In addition, as cortical magnification increases up the visual hierarchy, this estimate provides a lower bound for the proportion of cortical surface stimulated by our stimulus (*Harvey and Dumoulin, 2011*). We opted for this relatively small stimulus size as increasing the stimulus size also increases bar step size (assuming identical stimulation time). This in turn would have decreased the spatial precision at which (changes in) pRF parameters could be determined. Throughout the results section, we use the terms 'parafoveal' and 'peripheral' to indicate positions relative to the stimulated portion of visual field.

   The bar stimulus was composed of 1500 Gabor elements (4.34 cycle/dva spatial frequency, 9 arcmin sd, average luminance of 60 cd/m$^2$) projected over a dark-gray background (15 cd/m$^2$). Three times per bar location (i.e. every 533 ms), Gabor element parameters were updated to a new random location (uniformly distributed over the spatial extent of the bar at full width), a random orientation (uniformly drawn between 0 and 360°), a random color combination (either blue/yellow, or green/magenta) and a random new temporal frequency (TF; either high or low). We opted for color pairs that were opposite in terms of color opponency (*Hering, 1874*). This ensured that the Gabor peaks and troughs were made up of colors that provided strong color contrast. In addition, this meant that we presented stimuli to all color channels, in effect serving as a potent and full-range color stimulus. The high and low temporal frequencies were chosen per participant to facilitate their ability to distinguish TF changes (6 and 4 Hz in three participants, 7 and 3 Hz in two participants). The overall color and/or TF composition of the bar was transiently altered on some of these parameter updates, by changing the ratio of Gabor elements assigned either color combination or either TF (as targets for the behavioral tasks, see below). The temporal predictability of these events was minimized by randomly drawing occurrences according to an exponential distribution (mean 4 s, minimum 2 s). Additionally, the fixation mark central disk luminance either increased or decreased, with probability and duration of these occurrences identical to those of changes in the bar stimulus composition. These three types of transients (fixation mark luminance, bar color and TF composition) were independent, meaning they were randomly interleaved and could be combined on the screen. Importantly, this design ensured that physical stimulation was equated across all three attention conditions, which we describe below.

## Attention-PRF mapping task

For an overview of the stimulus and behavioral task see *Figure 1A*. Before each bar pass an automated voice (Apple OSX Dictation voice 'Kathy') informed participants to perform a two-alternative forced-choice task (2AFC) on one of the three stimulus parameter deviations. Task-relevant visual stimulus changes were accompanied by an auditory pure tone (440 Hz). This auditory cue alerted the participant to respond, while task-irrelevant stimulus changes occurred independently and without warning tone. This ensured that all task-related information was conveyed to the participant by auditory means, without concurrent changes in visual stimulation. The different stimulus changes (i.e. color, TF and fixation luminance) occurred independently and thus sometimes simultaneously, meaning the auditory tone was not reliably predictive of the stimulus dimension to attend. Therefore, participants needed to stably maintain condition-specific top-down attention throughout the duration of a bar pass. In the *Attend Color* condition, participants judged the relative predominance of blue/yellow or green/magenta Gabor elements in the bar stimulus, whereas in the *Attend TF* condition, participants judged the relative predominance of high compared to low TF Gabor elements in the bar stimulus. In the *Attend Fixation* condition, participants judged whether the central disk of the fixation mark increased or decreased in luminance. The magnitude of the stimulus changes was titrated by means of a Quest staircase procedure (*Watson and Pelli, 1983*), set to approximate 83% correct performance. To equate task difficulty across not only conditions but also bar stimulus eccentricity, we used separate Quest staircases at three different bar stimulus eccentricities in each of the attention conditions. Additionally, there was a separate staircase for the *Attend Fixation* task when no bar stimulus was on screen. This made for a total of 10 separate staircases during the experiment. Participants extensively practiced the task outside the scanner and staircases were reset before scanning. Each experimental run contained one bar pass per task condition, per direction, in random order (total of 24 bar passes per run).

## Feature preference and HRF mapper

We performed a separate randomized fast event-related fMRI experiment to (1) determine each voxel's relative preference for color and TF, and (2) to find the parameters that best described each participant's HRF, to be used in the pRF estimation procedure (see below). Full-field stimuli consisted of 8000 Gabor elements, uniformly distributed throughout the full circular aperture traversed by the pRF mapping stimulus ensuring identical density compared to the pRF mapping stimulus. Also, every 533 ms, all Gabor elements were assigned a new random orientation and location. These stimuli were presented for 3.2 s, with an inter-trial interval of 3.2 s. In a full factorial 2 × 2 design, we varied the color and TF content of the stimulus in an on-off fashion. That is, the TF of the Gabor elements was either 0 or 7 Hz, and the elements were either grayscale or colored (balanced blue/yellow and green/magenta). Trial order was determined based on an M-sequence (*Buracas and Boynton, 2002*), with no-stimulus (null) trials interspersed as a fifth category of trials. During this experiment, participants performed the same 2-AFC fixation-point luminance task as in the Attention-pRF Mapping Task (*Attend Fixation*), using a separate staircase. A single HRF was determined per participant using the R1-GLM approach (*Pedregosa et al., 2015*) on data from all conditions. The median HRF from the 1000 most responsive voxels (highest beta-weights in the colored high TF condition) was used as the participant-specific HRF. Visualizing the anatomical distribution of these voxels predominantly returned locations within early visual cortex. This, combined with the fact that attention was directed away from the stimulus, confirms that HRF estimation was mainly based on bottom-up color- and temporal frequency-driven visual responses.

## Data analysis

### MRI preprocessing

T1-weighted images were first segmented automatically using Freesurfer, after which the pial and grey/white matter surfaces were hand-edited. Regions of interest (ROIs) were defined on surface projected retinotopic maps using Freesurfer without the use of spatial smoothing. For every participant, one session's EPI image was selected as the target EPI, which was registered to his/her Freesurfer segmented T1-weighted image using the bbregister utility, after which the registration was hand-adjusted. Then, all EPI images were first motion corrected to their middle volume using FSL (*Jenkinson et al., 2012*) MCFLIRT to correct for within-run motion. Then, all EPI images were

registered both linearly (using FLIRT) and non-linearly (using FNIRT) to the mean-motion corrected target EPI to correct for between-run and session motion and inhomogeneities in B0 field. Low-frequency drifts were removed using a third order Savitzky-Golay filter (*Savitzky and Golay, 1964*) with a window length of 120 s. Arbitrary BOLD units were converted to percent-signal change on a per-run basis.

## pRF fitting procedure

pRF fitting and (statistical) parameter processing was performed using custom-written python pipeline (*van Es and Knapen, 2018*; copy archived at https://github.com/eLifeProduction/pRF_attention_analysis). Links to the data files required to reproduce the figures and analyses can be found in the readme of this repository. The fitting routines relied heavily on the scipy and numpy packages. We approximated the pRF by a two-dimensional isotropic Gaussian function. For an overview of our pRF fitting procedure see *Figure 1B*. A predicted time course for a given Gaussian function can be created by first computing the overlap of this function with a model of the stimulus for each time-point, and then convolving this overlap with the participant-specific HRF (*Dumoulin and Wandell, 2008*). It is possible to find these Gaussian parameter estimates using a minimization algorithm, but such an approach is at risk of missing the global optimum when parameters are not initialized at appropriate values. Recently, a model-free reverse-correlation-like method was developed, generating a pRF spatial profile without requiring any pre-set parameters (for details see *Lee et al., 2013*). Briefly, we employed this method using L2 regularized (Ridge) regression on a participant-specific-HRF convolved design matrix coding the stimulus position in a 31x31 grid for each timepoint, predicting data from all attention conditions together. Using a high regularization parameter ($\lambda = 10^6$), we used this procedure not to maximize explained signal variance, but to robustly determine the pRF center, which was defined as the position of the maximum weight. Having determined these approximate initial values for the pRF center, we next initialized a minimization procedure (*Powell (1964)* algorithm) at these location values, fitting position $(x, y)$, size, baseline and amplitude parameters of an isotropic 2D Gaussian to data from all conditions together using a design matrix with size 101x101 for enhanced precision. Then, all resulting Gaussian parameters were used to initialize a second minimization procedure which fitted a Gaussian for each attention condition separately at the same time (all parameters independent except for one shared baseline parameter). This approach allowed us to recover fine-grained differences in pRF parameters under conditions of differential attention.

## pRF selection

We discarded pRFs that were either at the edge of the stimulus region (above 3.3 dva in the *Attend Fixation* condition), or had size (standard deviation) larger than our stimulus diameter (7.2 dva) in any of the tasks (see *Figure 2—figure supplement 2* for details on the number of voxels that were rejected based on these criteria). Additionally, each voxel's contribution to all analyses was weighted according to the quality of fit of the pRF model, which was defined as 1 minus the ratio of residual to observed variance:

$$R^2 = 1 - \frac{\sum_i (m_i - p_i)^2}{\sum_i (m_i - \bar{m}_i)^2}$$

where $i$, $m$ and $p$ refer to voxel index, measured BOLD time-course and predicted time-course, respectively. We disregarded voxels with an $R^2 < .1$.

## pRF parameter analyses

Our sample size and statistical procedure were based on two recent and comparable studies (*Klein et al., 2014*; *Kay et al., 2015*) that adopted a 'dense sampling of individual brain approach' (*Poldrack, 2017*). This approach favors careful measurement of individual brains at the expense of large sample sizes in terms of the number of subjects. After defining visual ROIs per participant using standard retinotopic mapping procedures (*Dumoulin and Wandell, 2008*), this entailed pooling voxels for each ROI across participants. These results are presented in the main text and in *Supplementary file 1* -Tables 1-10. To verify that resulting effects are not driven by single

participants, we also performed all analyses for each participant separately, the results of which can be found in the figure supplements and in *Supplementary file 1* -Tables 11-26. Within the figure supplements, we refer to the pooling of voxels across subjects as 'super subject', to the analyses of individual subjects as 'per subject' and to the average across subjects as 'over subjects'.

When analyses were computed across voxels (cf. *Figures 2*, *3*, *4*, *6* and *7* and in the 'super subject' and 'per subject' methods in the figure supplements), p-values and confidence intervals were computed using $10^5$ fold bootstrap procedures. To test whether bootstrapped distributions differed from a certain threshold, p-values were defined as the ratio of bootstrap samples below versus above that threshold multiplied by 2 (all reported p-values are two-tailed). In the 'super subject' method, resulting p-values were corrected for multiple comparisons across ROIs using False Discovery Rate (FDR). When analyses were computed across participants, (cf. *Figures 5*, *8* and *9* and the 'over subjects' method in the figure supplements), statistics were performed using ANOVAs and/or t-tests. Our rationale for using bootstrapping across voxels and more classical tests across subjects is that while bootstrapping closely represents the actual distribution of the data, it is not stable for N < 20. We therefore made the additional assumption of normality when computing statistics across subjects.

The computation of the attentional modulation index (see below) included a division operation. This meant that when the denominator was very close to 0, this modulation index resulted in very extreme values. To prevent these extreme values from obfuscating general tendencies, we used a conservative threshold of five two-sided median absolute deviations. For reasons of consistency, we used this same outlier rejection procedure for all other analyses (resulting in slightly different N per analysis, see *Supplementary file 1* - Tables). Importantly, in all of these remaining analyses, outlier rejection did not meaningfully alter any result. Outlier rejection was performed per visual field bin (cf. *Figures 2B*, *3B/D*, *4A/B* and *6A/B/C*, scatters in *Figure 7*), per percentile bin (cf. *Figure 4C*), or per ROI (cf. 3C, 6D/E, correlations in *Figure 7*). When analyses were performed across participants, outliers were again rejected at the voxel level per participant, but all participants were always included (cf. *Figure 5A/B/C/D/E/F and and 8*). When comparing correlations to 0, correlations were Fisher transformed using the inverse hyperbolic tangent function.

## Feature attention modulation index

We computed a per-voxel index to quantify how strongly feature-based attention modulated the effects of spatial attention on spatial sampling (feature-based attention modulation index, or feature AMI). This measure combined pRF eccentricity and size parameters, as our results showed that spatial attention affected these parameters in concert (see *Figure 4*). Per voxel and per attention condition to the bar stimulus (*Attend Color* and *Attend TF*), we set up a two-dimensional vector containing difference in pRF eccentricity and size relative to the *Attend Fixation* condition. To ensure that pRF size and eccentricity contributed equally to the feature AMI, differences in pRF parameters in the *Attend Color* and *Attend TF* conditions relative to the *Attend Fixation* condition were normalized by the variance in the difference between the *Attend Stimulus* and *Attend Fixation* conditions. We then computed a feature AMI by dividing the difference between the norms of these vectors by their sum. This way, positive values of feature AMI indicate greater spatial attention effects on pRF parameters in the *Attend Color* condition than in the *Attend TF* condition and vice versa. Note that this measure quantifies the effects of feature-based attention regardless of the affected pRF parameter (i.e. eccentricity and size) and the sign of these changes (i.e. shifts toward or away from the fovea and increases or decreases in size).

## Attentional gain field modeling

To provide a parsimonious mechanistic account of how attention to the moving bar stimulus changed pRF position, we adapted an existing attentional gain model of attention (*Womelsdorf et al., 2008*; *Reynolds and Heeger, 2009*; *Klein et al., 2014*). This model conceptualizes the measured Gaussian pRF as the multiplication between a Gaussian stimulus-driven pRF (*SD*, i.e. the pRF outside the influence of attention), and a Gaussian attentional gain field (*AF*). Following the properties of Gaussian multiplication, the narrower the AF the stronger the influence on the SD, and the narrower the SD the smaller the resulting shift. An overview of model mechanics is shown in *Figure 10*. We estimated the SD by dividing the measured *Attend Fixation* pRF by an AF at fixation

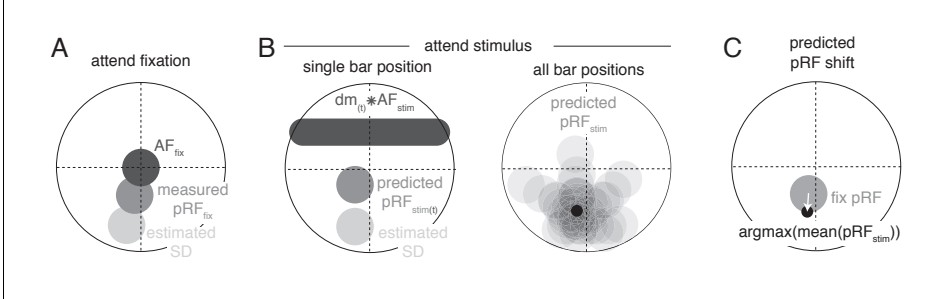

**Figure 10.** Schematic overview of modeling procedure. (**A**) The Stimulus Drive (SD) was estimated by dividing the measured Attend Fixation pRF by an AF at fixation ($AF_{fix}$). (**B**) Attention toward the bar stimulus at a given timepoint $t$ was modeled as the multiplication of the estimated SD with the bar stimulus at that timepoint ($dm_{(t)}$) convolved with another AF ($AF_{stim}$). These predicted Attend Stimulus pRFs were averaged over all timepoints. The maximum position of this profile was taken as the predicted Attend Stimulus pRF position. (**C**) The predicted pRF shift ran from the measured Attend Fixation pRF toward the predicted Attend Stimulus position.
DOI: https://doi.org/10.7554/eLife.36928.027

($AF_{fix}$, *Figure 10A*). As attention to the stimulus shifted the locus of attention as the bar moved across the screen, we modeled the effect of attention for each bar stimulus position separately (*Figure 10B*). Each unique bar stimulus (24 bar positions for each of 8 directions) was convolved with a Gaussian kernel ($AF_{stim}$) and multiplied with the estimated SD. This yielded one predicted pRF per bar position. These predicted pRFs were then scaled to a maximum of 1 and averaged over the different bar positions. This averaged profile essentially represented the pRF 'smeared' across visual space as spatial attention moved along with the bar stimulus throughout the recording. The peak of this averaged profile was then designated as the predicted pRF center location in the *Attend Stimulus* condition (*Figure 10C*). Thus, the modeling procedure consisted of two separate AFs, one at fixation ($AF_{fix}$) and one convolved with the bar stimulus ($AF_{stim}$), which were estimated at the same time. The input to the model was the *Attend Fixation* pRF, and the output was the predicted position for the *Attend Stimulus* pRF. Formally, this is given by:

$$pRF_{stim}pos = argmax\left( \frac{1}{n} \sum_{t=0}^{n} \left( \frac{\left(dm_{(t)}*AF_{stim}\right) \cdot \left(pRF_{fix}/AF_{fix}\right)}{max\left(\left(dm_{(t)}*AF_{stim}\right) \cdot \left(pRF_{fix}/AF_{fix}\right)\right)} \right) \right)$$

where $\left(dm_{(t)}*AF_{stim}\right)$ represents the stimulus design matrix at timepoint $t$ convolved with the AF toward the stimulus, $\left(pRF_{fix}/AF_{fix}\right)$ represents the estimation of the SD, and the denominator ensures scaling to a maximum of 1. To estimate how well a set of AF parameters fit the data across the entire visual field, we minimized the L2 distance between the predicted and measured *Attend Stimulus* pRF positions of the 64 vectors derived from the quadrant visual field format of *Figure 3B*. AF sizes were determined at an ROI level, thus assuming that attention influenced all pRFs within an ROI similarly, while possibly varying between ROIs. The model was evaluated for a 50 x 50 evenly spaced grid of AF sizes, where the AF at fixation varied between 1.5 and 2.5 dva, and the AF convolved with the stimulus varied between 0.6 and 1.6 dva (i.e. 0.02 dva precision). The convolution between the stimulus and the $AF_{stim}$ resulted in effective AF size to be 0.9 (bar stimulus width) larger than the $AF_{stim}$ itself. These parameter ranges therefore result in equal effective AF sizes. Reported sizes are the standard deviation of the 2D Gaussians, with 0.9 (the bar width) added to $AF_{stim}$ sizes. Modeling was performed for each participant separately.

## Gaze data processing

Gaze data was cleaned by linearly interpolating blinks detected by the EyeLink software. Transient occasions in which the tracker lost the pupil because of partial occlusion by the eyelid leading to high-frequency, high-amplitude signal components were detected and corrected as follows. Pupil size was first high-pass filtered at 10 Hz (the pupil impulse response function is a low-pass filter with a cutoff below 10 Hz (*Knapen et al., 2016*; *Korn and Bach, 2016*), after which those timepoints in which the acceleration of pupil size was greater than $10^5$ mm/s, and their neighbors within 5 s, were replaced with NaN values. Drift correction was performed within each bar-pass by subtracting the

median gaze position. All gaze positions were rotated to the direction of bar movement, after which we analyzed the median and variance (standard deviation) of the component in the direction of bar movement (i.e. the component relevant for the pRF measurement).

## Statistical tables file

All statistical Tables referenced in this manuscript can be found in a separate file attached to this submission, termed '*Supplementary file 1*'. These Tables specify the N, effect sizes and p-values for all ROIs and for the different statistical methods ('super subject', 'over subjects' and 'per subject'; see Materials and methods for definitions).

# Acknowledgements

This study was supported in part by an Open Research Area grant (ORA; #464-11-030) issued by the Netherlands Organization for Scientific Research (NWO) to JT.

# Additional information

### Funding

| Funder | Grant reference number | Author |
| --- | --- | --- |
| Nederlandse Organisatie voor Wetenschappelijk Onderzoek | ORA #464-11-030 | Jan Theeuwes |

The funders had no role in study design, data collection and interpretation, or the decision to submit the work for publication.

### Author contributions

Daniel Marten van Es, Conceptualization, Data curation, Software, Formal analysis, Investigation, Visualization, Methodology, Writing—original draft, Project administration, Writing—review and editing; Jan Theeuwes, Resources, Funding acquisition, Writing—original draft; Tomas Knapen, Conceptualization, Resources, Software, Supervision, Funding acquisition, Validation, Investigation, Methodology, Writing—original draft, Project administration, Writing—review and editing

### Author ORCIDs

Daniel Marten van Es (iD) http://orcid.org/0000-0002-7067-6394
Tomas Knapen (iD) http://orcid.org/0000-0001-5863-8689

### Ethics

Human subjects: Participant signed informed consent before participation in this study. The study was approved by the ethical review board of the University of Amsterdam (approval number 2016-BC-7145).

### Decision letter and Author response

Decision letter https://doi.org/10.7554/eLife.36928.033
Author response https://doi.org/10.7554/eLife.36928.034

# Additional files

### Supplementary files

• Supplementary file 1. Statistical tables.
DOI: https://doi.org/10.7554/eLife.36928.028

• Transparent reporting form
DOI: https://doi.org/10.7554/eLife.36928.029

## Data availability

All attention-pRF and feature-GLM results have been deposited on Figshare at https://doi.org/10.6084/m9.figshare.c.4012717.v1, under a Public Domain Dedication License. pRF fitting and (statistical) parameter processing code is available at https://github.com/daanvanes/PRF_attention_analysis (copy archived at https://github.com/eLifeProduction/pRF_attention_analysis).

The following dataset was generated:

| Author(s) | Year | Dataset title | Dataset URL | Database and Identifier |
|---|---|---|---|---|
| Daniel Marten van Es, Jan Theeuwes, Tomas Knapen | 2018 | pRF spatial and feature-based attention data | https://figshare.com/collections/PRF_attention_project/4012717/1 | figshare, 10.6084/m9.figshare.c.4012717.v1 |

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
