## [Decision Letter]

Thank you for submitting your article "Spatial sampling in human visual cortex is modulated by both spatial and feature-based attention" for consideration by *eLife*. Your article has been reviewed by two peer reviewers, and the evaluation has been overseen by a Reviewing Editor and Sabine Kastner as the Senior Editor. The following individual involved in review of your submission has agreed to reveal his identity: Michael Silver (Reviewer #2).

The reviewers have discussed the reviews with one another and the Reviewing Editor has drafted this decision to help you prepare a revised submission.

Summary:

This study measured effects of feature-based attention on pRF eccentricity and size changes related to spatial attention. The authors investigated the hypothesis that feature-based attention optimizes the attended feature's spatial sampling by using a pRF mapping stimulus that varied along two feature dimensions: color and temporal frequency, and they found that attention to color showed stronger modulation of pRF eccentricity and size change compared to attention to temporal frequency. The authors interpret these results with an attentional gain field model by noting that receptive fields preferring color are typically smaller and more foveal compared to those preferring temporal frequency, consistent with these receptive fields having greater changes in eccentricity. They further demonstrate that attentional gain fields were smaller for the *Attend Color* condition, indicating greater precision of the attentional gain field for color receptive fields. The editors were impressed with this elegant and well-written paper that contains thorough and principled analysis procedures. However, they were concerned regarding some of the statistical analyses (see details below). Also, the authors should clarify how the results advance the field's knowledge about effects of spatial and feature-based attention (and their interaction). The application of the attentional gain field model to these findings was perceived as a strength. Overall, the editors felt that major revisions will be necessary to strengthen the study, so that it can be further pursued at *eLife*.

Essential revisions:

1) Feature-based attention has long been thought to have modulatory effects on spatial attention (Saenz, Buračas and Boynton, 2002; Maunsell and Treue, 2006). The current study supports this idea by reporting specific analyses where color and temporal frequency are used as attended features and spatial resampling is assessed by measuring populations receptive fields. Despite the effort, the findings of the study are mostly confirmatory in nature. The manuscript would benefit from a reorganized presentation of the results to highlight the novel contributions of the study as opposed to confirmation of prior findings.

2) There isn't adequate information about how the target features were selected. Why should we be interested in attending to temporal frequency versus color in the context of this study? (It is also unclear/unexplained why blue-yellow and cyan-magenta were selected as representative colors.) Would anything change if different colors, a broader range of temporal frequencies, or entirely different feature sets (e.g., orientation) were used?

3) It is reported that pRF changes for *Attend Color* case are larger compared to *Attend Temporal Frequency* case. It would, however, be better if the difference could be presented in a more conspicuous way. For instance, the way it is presented in Figure 5A gives the impression that the difference is almost negligible. If there is merely a slight difference, more detailed discussion is warranted regarding the reasons underlying pRF changes when attending to color vs. temporal frequency.

4) Related to the previous comment: When the regions primarily engaged by the two distinct features compared lie at different points in the hierarchy of visual cortex, the strength of retinotopic representations (and thus spatial) will inevitably vary. Wouldn't that introduce a bias in assessment of interactions between spatial and feature-based attention?

5) There is too much variability across subjects in the attentional gain field sizes of temporal frequency vs. color (Figure 7). Given that there appears to be poor consistency across subjects for this measure, this figure does not convey information in a convincing way.

6) The authors combine pRF eccentricity and size to obtain a single index, because they claim there's a high correlation between the two. We don't think this is warranted. For instance, Table 22 shows that there's a significant correlation between eccentricity and size changes in V1 for only 2 out of 5 subjects, and the average correlation is not very high for this ROI either (R=0.461). This also makes it questionable to present the results using voxels from a "combined ROI", given the large degree of variability across individual ROIs.

7) In most of the supplementary figures (e.g., the first one), results for individual subjects are not shown at the same scale across subjects, which makes them confusing. Also, there's barely any change in the pRF size with increasing eccentricity for some ROIs of some subjects (e.g., V1 of s5). Some extended treatment and discussion of the consistency of the presented results across subjects would greatly benefit the manuscript.

8) We found it difficult that when statistical analyses are performed across voxels bootstrap tests were applied, and yet across subject tests were based on ANOVA or t-tests. If the same type of metric is evaluated in both cases, shouldn't you make the same assumptions about the distribution of the metric?

9) More detailed description of the outlier rejection method is needed, as this changes the set of voxels over which the results are reported. This procedure can potentially introduce biases within and across analyses, so it must be motivated very clearly. It is uncommon to reject voxels like this. If the goal is to report robust estimates of central tendency and variance of parameter estimates, then why not calculate them based on median and IQR?

10) It is stated that explained variance during pRF estimation is used to weigh the contributions of individual voxels to estimated parameters. This is understandable, but we would have expected to see actual reports of the explained variance across ROIs as well. This could help readers to judge and interpret the differences observed across ROIs.

11) The stimulus aperture size comprises a very small proportion of the visual field and is also much smaller than those used in other fMRI studies of attentional modulation of pRFs. The authors should provide more discussion about this issue, particularly with respect to the generalizability of their findings. Also, it seems problematic to make the distinction between parafoveal and peripheral pRFs, given that the locations of these pRFs only differ by a couple degrees of visual angle.

12) Figure 8 shows that behavioral accuracy (proportion correct trials) was approximately 80% for the different tasks and eccentricity bins. However, a Quest procedure was used to adaptively change stimulus parameters to generate target performance levels of 83%. Thus, the finding that percent correct did not vary across tasks and eccentricity thresholds is a trivial consequence of the adaptive psychophysical procedure, and statistical analysis providing evidence for the null hypothesis is relatively meaningless. While it would be difficult to directly compare task difficulty across the three tasks, the authors could compute psychophysical thresholds for each task and test whether these are influenced by eccentricity.

---

## [Author Response]

Essential revisions:1) Feature-based attention has long been thought to have modulatory effects on spatial attention (Saenz, Buračas and Boynton, 2002; Maunsell and Treue, 2006). The current study supports this idea by reporting specific analyses where color and temporal frequency are used as attended features and spatial resampling is assessed by measuring populations receptive fields. Despite the effort, the findings of the study are mostly confirmatory in nature. The manuscript would benefit from a reorganized presentation of the results to highlight the novel contributions of the study as opposed to confirmation of prior findings.

We would like to thank the reviewers for pointing out the necessity for better separating our confirmatory from our novel contributions. This comment has prompted us to rewrite the Introduction, Results and Discussion sections of the manuscript. The interaction between spatial and feature-based attention has indeed received generous scientific attention. Nevertheless, it remains elusive whether feature-based attention can influence spatial resampling that results from spatial attention. Although this relationship was predicted by behavioral work (Yeshurun and Carrasco, 1998, 2000; Yeshurun et al., 2008; Barbot and Carrasco, 2017), it has remained untested in electrophysiological recordings or functional neuroimaging. We are the first to directly assess this interaction in the present manuscript. In order to appreciate this novel contribution, we argue that it is of great importance to first present readers with a detailed description of the spatial resampling that resulted from spatial attention (which is mainly of confirmatory nature). Then, we can address how this is modulated by feature based attention. For this reason, we did not consider it opportune to change the order of presentation within the Results section. Instead, we implemented various changes to the text in order to better manage reader's expectations. For this, we first more concretely stress the novelty of our study in the Introduction:

“The studies mentioned above investigated modulatory effects of feature-based attention on spatial attention by measuring changes in response amplitude (e.g. ERP/firing rate). […] In the current study, we put this hypothesis to the test by measuring the brain's representation of space under conditions of differential attention.”

In addition, we added a paragraph in the Results section to reiterate this principal and novel aim, and to manage the reader's expectations about confirmatory vs. novel results:

“Our principal aim is to understand whether feature-based attention influences spatial resampling induced by spatial attention. […] Finally, we relate these feature-based attentional modulations to (1) bottom-up feature preference and (2) to differences in the spatial scale at which color and temporal frequency are processed.”

Finally, we also changed the order of the Discussion such that we now first discuss our novel findings, before we move on to relating our more confirmatory findings to the literature.

2) There isn't adequate information about how the target features were selected. Why should we be interested in attending to temporal frequency versus color in the context of this study? (It is also unclear/unexplained why blue-yellow and cyan-magenta were selected as representative colors.) Would anything change if different colors, a broader range of temporal frequencies, or entirely different feature sets (e.g., orientation) were used?

We will first clarify our reasoning for opting for color and temporal frequency. Next, we will argue why our results should not be specific for the specific features used in our experiment.

There are two important reasons for why we studied attending color versus temporal frequency. First, behavioral work suggested that the spatial scale at which features are processed should determine the degree of spatial resampling (Yeshurun and Carrasco, 1998, 2000; Yeshurun et al., 2008; Barbot and Carrasco, 2017). We therefore opted to use features that are known to be processed at different spatial scales. We previously only stressed a single factor that contributes such differential spatial sampling between these features: the eccentricity dependence of both pRF size and relative preference for color compared to temporal frequency. However, the cortical areas that preferentially process color and temporal frequency (i.e. MT+ compared to hV4; Liu and Wandell, 2005; Brouwer and Heeger, 2009, 2013; Winawer et al., 2010) differ in their average receptive field sizes (Amano et al., 2009; Winawer et al., 2010). Therefore, temporal frequency compared to color is processed on average by neurons with larger receptive fields. Finally, a third factor contributing to differential spatial sampling is that color compared to temporal frequency depends more on the parvocellular versus the magnocellular pathway, where spatial pooling is more fine-grained (Schiller and Malpeli, 1978; Hicks et al., 1983; Denison et al., 2014). In addition to this differential spatial sampling argument, the second main reason for opting for color and temporal frequency is that they are known to be processed in a distributed manner throughout the cortex (color: hV4 / temporal frequency: MT+). This provides an anatomically segregated handle for investigating the relationship between bottom-up feature preference and feature-based attentional modulations. As previously noted in our Materials and methods section, we specifically chose temporal frequency and not coherent motion, as coherent motion signals have been shown to influence pRF measurements (Harvey and Dumoulin, 2016). Also, it was previously shown that attention can be directed to these features (Wolfe and Horowitz, 2004; Cass et al., 2011). We now added a paragraph to our Introduction detailing this multi-faceted argument for why attending color and temporal frequency are of particular interest to our study:

“One important reason for studying the effects of attention to color and temporal frequency is that they are known to be processed at different spatial scales. […] Moreover, it was previously shown that attention can be directed to both feature domains (Wolfe and Horowitz, 2004; Cass et al., 2011).”

We also incorporated the two additional factors that contribute to differential spatial resampling between color and temporal frequency (i.e. parvocellular vs. magnocellular and offset in pRF size between hV4/MT+) in our Results section:

“What could then explain the fact that attending color in the stimulus induced greater changes in spatial sampling? […] In sum, the greater amount of spatial resampling when attending color can be parsimoniously explained by color being sampled by relatively smaller pRFs.”

And we discuss these two factors in our Discussion:

“Previous behavioral reports suggest that the spatial scale at which an attended feature is processed should influence the degree of spatial resampling (Yeshurun and Carrasco, 1998, 2000; Yeshurun et al., 2008; Barbot and Carrasco, 2017). […] Finally, both the current and previous studies show that pRF size (Dumoulin and Wandell, 2008) and color compared to temporal frequency preference (Curcio et al., 1990; Azzopardi et al., 1999; Brewer et al., 2005) vary across eccentricity such that foveal voxels have smaller pRFs and are more color sensitive.”

Previously, as mentioned above, we mainly emphasized the eccentricity gradient of both feature-preference and pRF size as underlying the finer spatial scale of color compared to temporal frequency processing. The initial version of our manuscript therefore contained a note about alternative eccentricity-dependent effects of attention. However, as we now stress two other factors that contribute to differences in spatial sampling between the two features, we currently argue that including this note leads to more distraction than that it is clarifying. We therefore removed these sentences from the Discussion:

“It must however be noted that as pRF size is closely related to pRF eccentricity, it is conceivable that some additional eccentricity dependent attentional modulation influenced our results. […] Therefore, although differential influences of attention across eccentricities have been observed before in the brain (e.g. Roberts et al., 2007; Bressler et al., 2013), these are likely brought about by differences in pRF size.”

Regarding the choice of specific colors, we would first like to note that after additional consideration we deem the name 'green' as more appropriate for the color previously indicated with 'cyan' (RGB color: (0,255,128)). We have therefore renamed the color pairs: blue/yellow and green/magenta. In our experiment, we opted for color pairs that were opposite in terms of color opponency (blue/yellow and green/magenta, Hering (1874)). This ensured that the Gabor peaks and troughs were made up of colors that provided strong color contrast. In addition, this meant that we presented stimuli to all color channels, in effect serving as a potent and full range color stimulus. We added these lines to the Materials and methods section:

“We opted for color pairs that were opposite in terms of color opponency (Hering (1874)). […] In addition, this meant that we presented stimuli to all color channels, in effect serving as a potent and full range color stimulus.”

Similarly, we used multiple temporal frequencies (3-7 Hz), in order to serve as a relatively general and broad temporal frequency stimulus. We argued above that it is the difference in spatial scale at which features are processed that influences the degree of spatial resampling. This implies that the particular choice of colors or temporal frequencies can only influence the degree of resampling if one has a particular reason for believing that different feature values are processed at different spatial scales. Attending these particular feature values in different attentional conditions should then lead to differential spatial resampling. We would like to point out however that in our experiment, attention was always devoted to all colors simultaneously within the *Attend Color* condition and to all temporal frequencies in the *Attend TF* condition. We therefore argue that our results reflect a general situation of attending color compared to temporal frequency. Finally, our results should indeed generalize to attending other feature dimensions (e.g. attending faces versus letters), as long as these features differ in the spatial scale of the visual system’s sensitivities to them. We added the abovementioned logic to the Discussion:

“In sum, we suggest that the greater degree of spatial resampling when attending color compared to temporal frequency can be explained by the difference in spatial scale at which these features are processed. […] This includes attending different feature values such as high compared to low spatial frequency, or attending different feature dimensions such as faces (broader spatial scale) versus letters (finer spatial scale).”

3) It is reported that pRF changes for Attend Color case are larger compared to Attend Temporal Frequency case. It would, however, be better if the difference could be presented in a more conspicuous way. For instance, the way it is presented in Figure 5A gives the impression that the difference is almost negligible. If there is merely a slight difference, more detailed discussion is warranted regarding the reasons underlying pRF changes when attending to color vs. temporal frequency.

Indeed, the absolute difference as presented in Figure 6A is relatively small. It should be noted however that this measure averages over the many changes in pRF parameters across attention to all possible bar positions. The magnitude of this measure should therefore not be interpreted as absolute shifts in degrees towards single attentional locations. A more interpretable measure is that of the feature based attentional modulation index (FAMI). This measure takes into account the average shift over all bar locations and computes feature based attentional modulations relative to this baseline. This FAMI measure is a contrast-like measure (difference divided by sum), and yields values around 0.05. This means that pRF changes were on average 10% larger when attending color compared to when attending temporal frequency. In fact, in some ROIs, the FAMI reaches values of 0.20, which roughly corresponds to 50% stronger pRF changes when attending color compared to temporal frequency. We would argue that an increase in pRF changes of 10-50% is not negligible and in fact is likely to have a substantial effect on visual information processing and perception. We added this logic to the Results section:

“Specifically, the feature AMI was around.05 on average across ROIs. As the feature AMI is a contrast measure where difference is divided by the sum, this roughly corresponds to 10% stronger pRF changes when attending color compared to temporal frequency. In some ROIs (V3AB/IPS0, see Figure 6—figure supplement 2), the FAMI reached values of.20, which roughly corresponds to 50% stronger pRF changes when attending color compared to temporal frequency.”

4) Related to the previous comment: When the regions primarily engaged by the two distinct features compared lie at different points in the hierarchy of visual cortex, the strength of retinotopic representations (and thus spatial) will inevitably vary. Wouldn't that introduce a bias in assessment of interactions between spatial and feature-based attention?

We agree that we did not sufficiently address the differences in visual hierarchy at which the two attended features are processed. In the current version of the manuscript, we now explicitly include this argument as one of the factors contributing to differences in the spatial scale at which the two attended features are processed (see our reply to Essential revision point 2). Indeed, it should be expected that offsets in pRF size should lead to differential pRF changes. Specifically, Gaussian interaction models of attention (Klein et al., 2014) suggest that the larger the stimulus drive (i.e. the pRF outside influence of attention), the greater the impact of attention. In correspondence, we observed that absolute pRF shifts were larger in areas with larger average pRFs (see Figures 3 and 4). Importantly however, our assessment of feature based attentional modulation is a contrast measure (AMI), relative to the absolute pRF modulation resulting from differential spatial attention. This analysis therefore controls for any offset in pRF size. Moreover, any difference in pRF size is also taken into account by the attentional gain field modeling (i.e. it is input to the model). Together, this means that differences in pRF size between visual areas that preferentially process the attended feature should not lead to biases in the assessment of interactions between spatial and feature based attention in our analyses. Nevertheless, (as we argue above) these offsets of pRF size between these regions does contribute to explaining why differential spatial resampling is required when attending color compared to temporal frequency.

5) There is too much variability across subjects in the attentional gain field sizes of temporal frequency vs. color (Figure 7). Given that there appears to be poor consistency across subjects for this measure, this figure does not convey information in a convincing way.

This comment has prompted us to change the presentation of our data in this figure for improved clarity (current Figure 8). Specifically, both previous findings (Klein et al., 2014) and our data (Figure 5) show that a single attentional gain field operates throughout visual cortex. In addition, we did not find that feature-based attentional modulation was related to feature preference across visual areas (Figure 6E). We therefore summarized feature-based attentional modulations of the attentional gain field across visual areas as the median across individually fitted ROIs (previously dark shaded area) and as the combined ROI (i.e. fitted on data from all ROIs together). We clarified this in our manuscript:

*“*Both our data (Figure 5) and previous findings (Klein et al., 2014) showed that a single attentional gain field affects the different visual regions similarly. […] We therefore analyzed feature-based attentional modulations of the attentional gain field both on data from all ROIs fitted together (the 'combined ROI'), and as the median across individually fitted ROIs.”

6) The authors combine pRF eccentricity and size to obtain a single index, because they claim there's a high correlation between the two. We don't think this is warranted. For instance, Table 22 shows that there's a significant correlation between eccentricity and size changes in V1 for only 2 out of 5 subjects, and the average correlation is not very high for this ROI either (R=0.461). This also makes it questionable to present the results using voxels from a "combined ROI", given the large degree of variability across individual ROIs.

In order to verify whether our feature-based attentional modulation (feature AMI) results were not specific to our combined measure of eccentricity and size changes, we repeated this analysis using pRF size and eccentricity separately and report them in the manuscript (see the novel Figure 6—figure supplement 3). This analysis returned highly similar results compared to our initial combined pRF size and eccentricity feature-based AMI. In our opinion, it is warranted to combine the pRF size and eccentricity differences into a single measure of attentional modulation. We added a reference to this figure in the Results section:

“Computing the AMI with either pRF eccentricity or size changes separately (i.e. not as a combined measure) returned similar results (see Figure 6—figure supplement 3).”

We agree that it is striking that the correlation between eccentricity and size is somewhat lower in V1 compared to the other ROIs. We suspect that this might be due to the fact that pRFs are very small in V1. This in turn means that absolute attention-induced changes in pRF eccentricity and size are also very small and therefore harder to measure with high precision. In addition, we showed above that feature-based attentional pRF modulations are present for both eccentricity and size changes separately. So, although these correlations are smaller in V1, we conclude it is likely that this is not the result of fundamental differences in the effects of attention. We continue to feel that it is meaningful to combine all voxels into a single 'combined' ROI, as it serves to show the predominant effects present across all voxels in the visual brain. We want to stress that the combined ROI is not meant to present results averaged across ROI. Indeed, the different ROIs contribute meaningfully differently to this combined ROI (also see Essential revision point 10). The individual ROI results then serve to specify this overall effect per ROI. In our view, the two analyses serve complementary purposes. For these reasons, we believe that including the combined ROI helps the reader to better appreciate the predominant patterns in the data.

7) In most of the supplementary figures (e.g., the first one), results for individual subjects are not shown at the same scale across subjects, which makes them confusing. Also, there's barely any change in the pRF size with increasing eccentricity for some ROIs of some subjects (e.g., V1 of s5). Some extended treatment and discussion of the consistency of the presented results across subjects would greatly benefit the manuscript.

In order to provide additional assessment of the stability of the results across subjects, we matched the scale of all the individual subject figures. In addition, we performed additional analyses (see novel Figure 2—figure supplement 1, and Tables 1 and 11) on the slope between eccentricity and size in all ROIs and in all individual subjects. Although this relationship was numerically positive in all ROIs in all subjects, it did not cross the significance threshold of α=.05 in some ROIs (VO/V3AB/IPS0/MT+; p’s of. 146,. 073,. 078 and. 058 respectively) in no more than 1 out of 5 subjects. We therefore removed the incorrect 'in all participants' statement from the main text and apologize for its initial inclusion. In order to provide more extended discussion of the results across subjects, we added a novel paragraph to the Results section:

“In order to evaluate the stability of our results we repeated all analyses for individual subjects. […] In addition, these analyses showed that spatial resampling was consistently modulated by feature-based attention across subjects (i.e. feature AMI was on average 0.059 greater than 0 (F_(1,4)_ = 18.868, p =.012, η2p=.394), and was not different between ROIs (F_(8,32)_ = 0.631, p =.746, η2p=.066)).”

8) We found it difficult that when statistical analyses are performed across voxels bootstrap tests were applied, and yet across subject tests were based on ANOVA or t-tests. If the same type of metric is evaluated in both cases, shouldn't you make the same assumptions about the distribution of the metric?

Our rationale for using bootstrapping across voxels is that this allowed for the computation of confidence intervals for metrics that can only be derived across a set of voxels (e.g. correlation values). This also allowed us to visualize the distribution of the metric in a data-driven fashion. Letting go of the assumption of normality in bootstrapping thereby increased the specificity of the derived results. We would preferably also have used bootstrapping when calculating statistics over subjects, but bootstrapping is not stable for N<20. We made the additional assumption of normality for across subjects in order to increase stability and robustness of the derived p-values. We thus utilized both bootstrapping and more classical statistics to optimize the balance between sensitivity and robustness. We added these sentences to the Materials and methods section in order to clarify this issue:

“Our rationale for using bootstrapping across voxels and more classical tests across subjects is that while bootstrapping closely represents the actual distribution of the data, it is not stable for N < 20. We therefore made the additional assumption of normality when computing statistics across subjects.”

In order to complete the consistency of using bootstrapping across participants and classical statistics across voxels, we now changed the visualization of confidence intervals in Figure 9A from using bootstrapping to classical statistics. Note that this is merely a matter of visualization, as the Bayesian analyses reported are not affected. Updated CIs are slightly wider, as classical statistics is more conservative than bootstrapping for small Ns.

9) More detailed description of the outlier rejection method is needed, as this changes the set of voxels over which the results are reported. This procedure can potentially introduce biases within and across analyses, so it must be motivated very clearly. It is uncommon to reject voxels like this. If the goal is to report robust estimates of central tendency and variance of parameter estimates, then why not calculate them based on median and IQR?

In computing the AMI, division by values close to zero sometimes resulted in extreme values. In order to remove these values from further analysis, we opted to use a very conservative outlier rejection method, using 5 median absolute deviations as a threshold. We specifically opted to apply the same outlier rejection method to all other analyses for consistency. In order to verify whether outlier rejection had any significant effect on the other analyses, we also repeated all analyses without outlier rejection and found no meaningful changes. Using the median and IQR, as the reviewers suggest, would have made for a suitable alternative. This should have yielded very similar results, as our method similarly rejected outliers based on deviations from the median. Indeed, our approach changed the set of voxels over which the results are reported. Yet, the IQR method would similarly base its results on different subpopulations of voxels per analysis (namely those voxels not too far from the median), without explicitly stating so in changes in N. We therefore argue that our approach is both conservative, consistent (across analyses) and transparent (stating the N). We've added a more detailed description of this outlier rejection method in the Materials and methods section:

“The computation of the attentional modulation index (see below) included a division operation. […] Importantly, in all of these remaining analyses, outlier rejection did not meaningfully alter any result.”

In order to further increase the consistency of our outlier rejection method, we simplified the determination of outliers in the feature-AMI analysis. Previously, outliers were determined based on multiple measures, including the color AMI, the TF AMI, the combined AMI and on feature-preference. Currently, we only determine outliers based on the feature-AMI in this analysis. This increased the number of included voxels and slightly changed the exact statistical values in the feature-AMI analyses (Tables 9 and 25 and legend of Figure 6—figure supplement 4). In maximally 1/5 subjects, this resulted in changes of statistical significance (in V3, LO, and MT+). Specifically, in V3 this resulted in one additional subject showing significantly positive AMI; in MT+ it resulted in 1 fewer subject showing significantly positive AMI; in LO, it resulted in 1 fewer subject showing significantly negative AMI. As a result, none of the subjects show a significantly negative AMI in any ROI. In addition, it did not change any of the conclusions about statistical significance for the ‘super subject’ method nor when computing t- and F-tests over subject values (‘over subjects’ method).

10) It is stated that explained variance during pRF estimation is used to weigh the contributions of individual voxels to estimated parameters. This is understandable, but we would have expected to see actual reports of the explained variance across ROIs as well. This could help readers to judge and interpret the differences observed across ROIs.

We agree. In order to present additional information regarding the distribution of explained variance across the different ROIs, we added two panels to Figure 2 (and added Figure 2—figure supplement 2). The first panel (C) depicts visualizations of the distributions of explained variance across voxels for each ROI. The second panel (D) displays visualizations of the number of voxels per ROI. Combining the information from these two figures provides the reader with a comprehensive overview of the total amount of signal that was present in each ROI. In addition, we added a supplementary figure that details this information per subject, and also shows how many voxels remain in each ROI after the application of the different rejection criteria.

11) The stimulus aperture size comprises a very small proportion of the visual field and is also much smaller than those used in other fMRI studies of attentional modulation of pRFs. The authors should provide more discussion about this issue, particularly with respect to the generalizability of their findings. Also, it seems problematic to make the distinction between parafoveal and peripheral pRFs, given that the locations of these pRFs only differ by a couple degrees of visual angle.

We understand that our stimulus could be considered small compared to the number of degrees of visual angle that spans our visual world (about 5% assuming full visual field of 144 degrees). Even though our stimulus only spanned roughly 5% of the visual field, this corresponds to approximately 25% of V1 surface due to cortical magnification (Adams and Horton, 2003). As this bias for over representing the central part of the visual field increases up the visual hierarchy (Harvey and Dumoulin, 2011), this percentage represents a minimum bound of cortical surface area stimulated by our stimulus. Comparing our stimulus size to other studies of attention and pRFs, we find a maximum stimulus eccentricity of 4.5 degrees in Vo et al., 2017, (corresponding to roughly 30% of V1 surface as approximated based on the results by Adams and Horton, 2003), of 5 degrees eccentricity in Klein et al., 2014, (~33% V1 surface), of 7.5 degrees in Kay et al., 2015, (~45% of V1 surface), and of 14 degrees in Sheremata and Silver, 2015, (~60% of V1 surface). Although our stimulus size is thus relatively small, it is generally comparable to two of these studies in terms of proportion of stimulated V1 (25% in our study compared to 30% and 33% in Vo et al., 2017, and Klein et al., 2014, respectively). Yet, one might argue that 25% is still a low number, and that the stimulus preferably should have been larger. However, increasing stimulus size also increases bar stimulus step size (keeping stimulation time equal). This means that the precision at which pRF properties like position and size can be determined decreases with increasing stimulus size. As we sought to measure changes in spatial sampling with the highest possible fidelity, we opted to keep the stimulus size relatively small in order to maximize our sensitivity for measuring changes in pRF parameters. Finally, we understand that the term 'parafoveal' and 'peripheral' are not correct in absolute terms. We meant to use those terms as relative, rather than absolute measures of eccentricity. We now specifically added a section to clarify this in our Materials and methods section:

“Although our stimulus spanned roughly 5% of the visual field (assuming a visual field of 144 dva), this corresponds to roughly 25% of V1 surface area due to cortical magnification (Adams and Horton, 2003). […] Throughout the Results section, we use the terms 'parafoveal' and 'peripheral' to indicate positions relative to the stimulated portion of visual field.”

12) Figure 8 shows that behavioral accuracy (proportion correct trials) was approximately 80% for the different tasks and eccentricity bins. However, a Quest procedure was used to adaptively change stimulus parameters to generate target performance levels of 83%. Thus, the finding that percent correct did not vary across tasks and eccentricity thresholds is a trivial consequence of the adaptive psychophysical procedure, and statistical analysis providing evidence for the null hypothesis is relatively meaningless. While it would be difficult to directly compare task difficulty across the three tasks, the authors could compute psychophysical thresholds for each task and test whether these are influenced by eccentricity.

We agree with the reviewers that not finding any difference in task performance is to be expected using a Quest staircase procedure. Nevertheless, using this procedure does not necessarily guarantee success. It is for instance possible that the procedure settles at a wrong difficulty level, making this verification necessary. We would argue that Quest and similar procedures are too often used as a panacea without this necessary verification. The analysis presented in the manuscript therefore serve to verify the validity of the Quest procedure in achieving the desired level of accuracy. We changed the introductory text to this part of the Results section to more clearly state this aim:

“Although we used a Quest procedure to equate difficulty across attention conditions and across different levels of eccentricity, it is possible that this procedure stabilized at a faulty difficulty level. In order to verify whether the Quest procedure successfully equated performance we used a similar Bayesian approach, testing whether a model including attention condition (3 levels) and stimulus eccentricity (3 levels) influenced behavioral performance (Figure 9A).”

We would like to note that stimuli were the same across attention conditions, and that our procedure successfully equated task difficulty. These were the chief aims of our behavioral paradigm. Yet, this reviewer comment has prompted us to also analyze the ratio of Gabor elements of either feature value used by the Quest procedure in order to equate difficulty. We placed these analyses in a novel figure supplement (see Figure 9—figure supplement 1). The legend of this supplement reads as the following:

**“**Figure 9—figure supplement 1.Ratio of Gabor elements of either feature value used by the Quest procedure in order to equate difficulty. […] This could suggest that the greater degree of spatial resampling we observed when attending color discounted the lower relative sensitivity for color in the periphery.”